# Nuclear glycine decarboxylase suppresses STAT1-dependent MHC-I and promotes cancer immune evasion

Rui Liu, Hui-Fang Li, Qi Jiang, Jun-Ge Shi, Zi-Lun Ruan [ID], Peng Ren, Yi-Nuo Li, Hong-Bing Shu [ID] & Shu Li [ID] ✉

## Abstract

Inadequate antigen presentation by MHC-I in tumor microenvironment (TME) is a common immune escape mechanism. Here, we show that glycine decarboxylase (GLDC), a key enzyme in glycine metabolism, functions as an inhibitor of MHC-I expression in EGFR-activated tumor cells to induce immune escape by a mechanism independent of its enzymatic activity. Upon EGFR activation, GLDC is phosphorylated by SRC and subsequently translocated to the nucleus in human NSCLC cells. Nuclear GLDC sequesters STAT1 co-activator SMARCE1, inhibiting STAT1-dependent transcription of the inflammatory genes *IRF1* and *NLRC5*. Further, GLDC recruits DNMT1 to the *IRF1/NLRC5* promoter inducing DNA hypermethylation, suppressing transcription of downstream MHC-I genes. Inhibition of GLDC restores MHC-I levels in tumor cells, improves tumor-specific CD8[+] T cells functions in the TME, and rescues anti-tumor effects of PD-1 blockade therapy in mice. Our findings reveal a non-enzymatic nuclear function for GLDC in the suppression of MHC-I antigen presentation, suggesting new strategies for ICB-based combination immunotherapy.

**Key words** MHC-I Antigen Presentation; Immune Escape; GLDC; EGFR Activation; ICB Therapy
**Subject Categories** Cancer; Chromatin, Transcription & Genomics; Immunology

## Introduction

The advances of immune checkpoint blockade (ICB) therapy have revolutionized the treatment of the cancer landscape (Liu et al, 2024; Sharpe and Pauken, 2018; Waldman et al, 2020). Clinical ICB therapy, primarily anti-PD-1, anti-PD-L1, and anti-CTLA4, effectively restores tumor-specific T cell immunity by targeting general immune evasion mechanisms in the tumor microenvironment (TME), exhibiting remarkable objective response and durable remission across numerous tumor types (Waldman et al, 2020).

However, not all patients respond to ICB therapy, and those who initially respond may also develop resistance. Patients who respond to ICB therapy are generally considered to have "hot" tumors (also known as "inflamed" tumors) with abundant tumor-specific CD8[+] T cell responses, which are required for effective initiation of cytotoxic antitumor responses (Morad et al, 2021). The majority of tumors are immunologically "cold" and lack significant T cell infiltration, resulting in them being unresponsive to ICB therapy or to develop acquired resistance (Morad et al, 2021; Zhang et al, 2022). The causes that contribute to low immunogenicity of TME have been widely reported, such as loss of antigen presentation, low tumor neoantigen load and impaired IFNγ response (Zhang et al, 2022). Among these causes, inadequate antigen presentation leads to a hindrance of the T cell immune response, accounting for a large proportion of patients (Bell and Zou, 2024). These observations emphasize that increasing antigen presentation levels in TME is a promising strategy to improve the efficacy of ICB therapy.

The major histocompatibility complex class I (MHC-I) antigen presentation pathway determines the specificity of CD8[+] T cells and is critical for their activation and proliferation (Wang et al, 2024). The basic principle of MHC-I-mediated antigen presentation is that MHC-I presents intracellular peptide antigens to form stable peptide-MHC-I (pMHC-I) complexes on the cell surface, which is recognized by antigen-specific T cell receptor (TCR) of specific CD8[+] T cells, thereby triggering CD8[+] T cell responses (Rossjohn et al, 2015; Sykulev, 2023; Wang et al, 2024). Genetic mutations or loss of expression of the major components involved in antigen presentation pathway and processing, such as MHC-I and beta-2 microglobulin (B2M) (MHC-I complex), transporters associated with antigen processing 1 and 2 (TAP1/2), and Tapasin (TAPBP) (peptide transportation and loading), are the major causes of insufficient antigen presentation in the TME, inducing tumors to evade cytotoxic CD8[+] T cell responses and compromise the efficacy of ICB therapy (Bell and Zou, 2024; Sykulev, 2023). However, the specific reintroduction of these factors into TME may be difficult as a therapeutic strategy. Studies of the underlying mechanisms responsible for the inadequate antigen presentation in TME are important for the development of ICB-based combination immunotherapy strategies.

Several recent studies have revealed regulatory mechanisms of MHC-I. It has been shown that MHC-I is selectively degraded in lysosomes by an autophagy-dependent mechanism, and inhibition of autophagy

Department of Infectious Diseases, Medical Research Institute, Zhongnan Hospital of Wuhan University; Frontier Science Center for Immunology and Metabolism, Taikang Center for Life and Medical Sciences; Wuhan University, Wuhan 430071, China. ✉E-mail: shuli@whu.edu.cn

results in increased antigen presentation and synergies with ICB therapy (Yamamoto et al, 2020). Membrane-localized MHC-I is associated with the transmembrane proteins SUSD6 and TREM127, which recruit the E3 ubiquitin ligase WWP2 to ubiquitinate MHC-I, leading to its lysosomal degradation. Inhibition of this membrane-associated MHC-I degradation axis enhances antigen presentation and antitumor immunity (Chen et al, 2023). The cholesterol metabolism regulator protein PCSK9 can disrupt the recycling of MHC-I to cell surface by associating with it physically and promoting its relocation and degradation in the lysosome, inhibiting PCSK9 increases the expression of MHC-I and intratumorally infiltration of cytotoxic T cells (Liu et al, 2020). TRAF3, a suppressor of the NF-κB pathway, is identified as a negative regulator of MHC-I expression. The TRAF3-knockout signature is associated with higher MHC-I in primary patient samples and better ICB response (Gu et al, 2021). ACSL5 functions as an immune-dependent tumor suppressor. ACSL5 deficiency impairs MHC-I expression by inhibition of NLRC5 (Lai et al, 2024). However, these factors may not cover the causes of inadequate MHC-I antigen presentation in all tumor types, and the exact clinical therapeutic significance of these factors in modulating tumor-associated MHC-I antigen presentation needs to be further validated. With increasing evidence that the low numbers of tumor-specific T cells in TME are highly correlated with MHC-I levels, it is necessary to identify more cancer-specific regulators of MHC-I antigen presentation pathways as potential therapeutic targets to match tumor patients of different types, enhancing the function of tumor-specific CD8[+] T cells and the efficacy of ICB therapy (Scheper et al, 2019).

In addition to maintaining the normal physiological function of cells, metabolic enzymes have been discovered to be involved in tumor initiation, development, metastasis and immune escape (Hu et al, 2024; Lai et al, 2024; Liu et al, 2020; Wang et al, 2019; Yang et al, 2012). Cancer cells often reprogram metabolic enzymes to meet their own malignant needs. However, whether and how alterations of the metabolic enzymes regulate MHC-I-based tumor immunity remain largely unknown. In this study, we used a CRISPR screening method to identify the rate-limiting metabolic enzymes involved in MHC-I regulation. We found that glycine decarboxylase (GLDC), a key enzyme in glycine metabolism, is a functional MHC-I inhibitor in EGFR-activated tumor cells. Mechanistically, EGFR activation triggers SRC-mediated phosphorylation and nuclear translocation of GLDC. The nuclear GLDC hijacks the STAT1 co-activator SMARCE1 to inhibit the binding of STAT1 to promoter regions of *IRF1* and *NLRC5* genes, and the GLDC/SMARCE1/STAT1 complex also recruits DMAP1/DNMT1 to induce promoter DNA hypermethylation of *IRF1* and *NLRC5* genes, resulting in transcriptional inhibition of the downstream MHC-I genes. Inhibition of GLDC increases MHC-I levels in tumor cells and promotes antitumor immunity. Our findings reveal a non-enzymatic nuclear function of GLDC in the inhibition of MHC-I expression and antitumor immunity, and provide a foundation for designing new strategies to enhance cancer immunotherapy.

# Results

## GLDC inhibits MHC-I surface expression independently of its enzymatic activity

To identify whether metabolic enzymes exert inhibitory functions of the MHC-I antigen presentation pathway, we used the CRISPR-Cas9 method to individually knock out 107 rate-limiting metabolic enzymes (Appendix Table S1) in human H1299 non-small-cell lung carcinoma (NSCLC) cells, which exhibit a constitutive high level of MHC-I on the cell surface and high efficiency of sgRNA transduction in the cells, and then measured the effects on MHC-I expression on the cell surface. In this screen, B2M, a shared component of all MHC class I molecules and whose loss results in complete absence of MHC-I surface expression (Beck et al, 2024), was used as a positive control. Among the examined metabolic enzymes, knockout of four genes, including DDC, TYR, OGDHL and GLDC, most dramatically upregulated MHC-I expression on the cell surface (Fig. 1A). Subsequent validation experiments revealed that GLDC knockout produced the most pronounced increase in the surface expression of MHC-I (Appendix Fig. S1A). Notably, knockout of the other three genes significantly compromised cellular viability, suggesting their essential roles in fundamental metabolic processes. Therefore, we prioritized GLDC for further investigation. GLDC is a key enzyme of the glycine cleavage system that converts glycine into one-carbon units (Liu et al, 2021). It has been demonstrated that GLDC is hyperactive in different types of cancer cells and plays a fundamental role in tumor growth (Liu et al, 2022; Liu et al, 2021; Mukha et al, 2022; Zhang et al, 2012). However, its function in tumor immune evasion is unknown. GLDC-deficiency upregulated MHC-I surface expression in human A549, H1299 or PC9 NSCLC cells (Fig. 1B; Appendix Fig. S1B). In addition, GLDC-deficiency also increased MHC-I surface expression in mouse CT26 and MC38 colorectal carcinoma cells, which is constitutively expressing MHC-I (Appendix Fig. S1C,D). B16-OVA melanoma cells and Lewis Lung Carcinoma (LLC) cells express low levels of MHC-I, GLDC-deficiency increased IFNγ-induced expression of MHC-I in these cells (Appendix Fig. S1C,D). Taken together, these results suggest that GLDC is a negative regulator of MHC-I surface expression.

Deregulation of epidermal growth factor receptor (EGFR) signaling has a central role in driving cancer lung pathogenesis, including immune evasion (Friedlaender et al, 2022; Wang et al, 2019). A549 and H1299 cells harbor wild-type EGFR and constitutively active EGFR signaling, whereas PC9 cells harbor an activating mutation—a deletion of Glu746-Ala750 in exon 19 of the EGFR gene (EGFR[ex19del]). Treatment with Afatinib, a specific EGFR inhibitor, increased the surface expression of MHC-I in A549 and PC9 cells (Fig. 1C and Appendix Fig. S1E). Notably, EGFR inhibition impaired the upregulation of MHC-I induced by GLDC-deficiency (Fig. 1C and Appendix Fig. S1E). Next, we transduced the mouse tumor cells (CT26, MC38, B16-OVA, and LLC) with EGFR[ex19del] and analyzed MHC-I levels on the cell surface (Appendix Fig. S1F). The results indicated that overexpression of EGFR[ex19del] inhibited MHC-I surface expression, which was impaired by GLDC-deficiency (Fig. 1D and Appendix Fig. S1G). We further assessed antigen presentation by detecting the cell surface expression of H-2K[b]-bounded OVA-derived SIINFEKL peptide complex and found that GLDC-deficiency reversed EGFR[ex19del]-mediated downregulation of antigen presentation (Fig. 1E). These results suggest GLDC plays a contributory role in EGFR-mediated MHC-I suppression, and EGFR can also down-regulate MHC-I through GLDC-independent mechanisms. This observation aligns with previous studies reporting alternative pathways by which EGFR inhibits MHC-I antigen presentation (Brea et al, 2016; Wang et al, 2025; Watanabe et al, 2019). Next, we

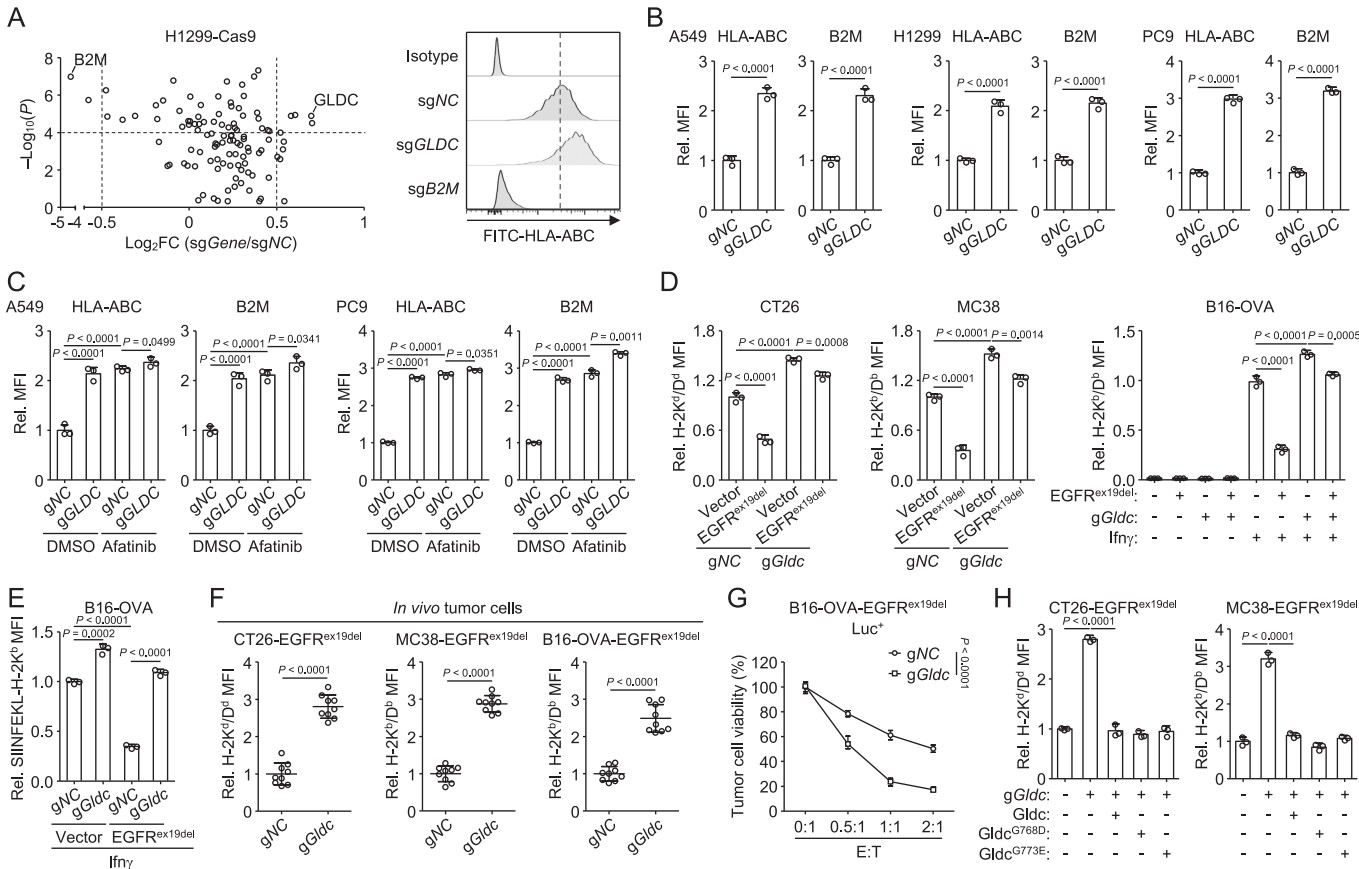

**Figure 1. GLDC inhibits MHC-I surface expression independently of its enzymatic activity.**

(A) Volcano plot showing the relative effects of knockout of the metabolic enzyme genes on MHC-I expression on the cell surface. H1299-Cas9 cells were transduced with sgRNAs targeting 107 metabolic enzyme genes (2 sgRNAs per metabolic enzyme gene). The cells were stained with FITC-HLA-ABC antibody and analyzed by flow cytometry. Data of the relative HLA-ABC MFI (sg*Gene*/sg*NC*) were shown as a volcano plot. The graph shows the mean, $n = 3$ independent samples. Data were analyzed using a student's unpaired *t*-test with GraphPad Prism 8. MFI, median fluorescence intensities. (B) GLDC-deficiency enhances MHC-I surface expression. The indicated cells were stained with the indicated antibodies and analyzed by flow cytometry. Graph shows mean ± SEM, $n = 3$ independent samples. Data were analyzed using a student's unpaired *t*-test with GraphPad Prism 8. (C) Afatinib treatment inhibits upregulation of MHC-I induced by GLDC-deficiency. The indicated cells were treated with DMSO or Afatinib ($2 \, \mu M$) for 24 h before flow cytometry analysis with the indicated antibodies. Graph shows mean ± SEM, $n = 3$ independent samples. Data were analyzed using two-way ANOVA with GraphPad Prism 8. (D, E) Effects of EGFR activation on MHC-I surface expression. CT26 and MC38 cells were stained with the indicated antibodies and analyzed by flow cytometry. B16-OVA cells were stimulated with Ifnγ (50 ng/mL) for 24 h before flow cytometry analysis with the indicated antibodies. Graph shows mean ± SEM, $n = 3$ independent samples. Data were analyzed using two-way ANOVA with GraphPad Prism 8. (F) Effects of GLDC-deficiency on MHC-I surface expression of tumor cells in vivo. The indicated cells were subcutaneously injected into mice. Tumor-bearing mice were euthanized on day 18 (CT26-EGFR[ex19del] tumor), day 12 (MC38-EGFR[ex19del] tumor) or day 14 (B16-OVA-EGFR[ex19del] tumor), and then the tumor tissues were separated from the mice. Tumor cells were isolated from the indicated tumor tissues, stained with the indicated antibodies and analyzed by flow cytometry. Graph shows mean ± SEM, $n = 9$ independent samples. Data were analyzed using a student's unpaired *t*-test with GraphPad Prism 8. (G) Effects of GLDC-deficiency on CD8[+] T cell-mediated cytotoxicity. The indicated B16-OVA-EGFR[ex19del] Luc[+] cells were cocultured with OT-I cells for 36 h at the indicated effector-to-tumor ratio (E:T) before luciferase assays. Graph shows mean ± SEM, $n = 4$ independent samples. Data were analyzed using two-way ANOVA with GraphPad Prism 8. (H) Effects of GLDC mutations on MHC-I surface expression. The indicated CT26-EGFR[ex19del] or MC38-EGFR[ex19del] cells were reconstituted with mouse wild-type Gldc, Gldc[G768D] or Gldc[G773E] and then analyzed by flow cytometry with the indicated antibodies. Graph shows mean ± SEM, $n = 3$ independent samples. Data were analyzed using two-way ANOVA with GraphPad Prism 8. Source data are available online for this figure.

examined the effects of GLDC-deficiency on MHC-I alloantigen levels on the surface of mouse tumor cells grown in vivo. We established eGFP-expressed CT26-EGFR[ex19del], MC38-EGFR[ex19del], or B16-OVA-EGFR[ex19del] cells, subcutaneously injected these cells into Balb/c (CT26-EGFR[ex19del] cells) or C57bl/6j (MC38-EGFR[ex19del] or B16-OVA-EGFR[ex19del] cells) mice and examined MHC-I expression on the eGFP[+] population from tumor tissue single-cell suspension of tumors. We observed that MHC-I surface expression was upregulated in GLDC-deficient eGFP[+] tumor cells compared to control eGFP[+] tumor cells (Fig. 1F), and tumor growth in mice

subcutaneously injected with control cells was markedly faster than those injected with GLDC-deficient cells (Appendix Fig. S2A).

Considering MHC-I antigen presentation is one of the major determinants that control CD8[+] T cell activation and function, we next investigated whether GLDC modulates the tumoral sensitivity to CD8[+] T cell-mediated cytotoxicity. We transduced a lentivector expressing luciferase (Luc) into B16-OVA-EGFR[ex19del] cells, then cocultured these tumor cells with activated OVA-specific CD8[+] T (OT-I) cells and evaluated the tumor viability by quantifying tumoral luciferase activities. The viability of tumor cells was

gradually reduced along with the increased effector-to-target (E:T) ratio (Fig. 1G). However, at the same E:T ratio, the viability of GLDC-deficient B16-OVA-EGFR$^{ex19del}$ cells were much lower than control B16-OVA-EGFR$^{ex19del}$ cells (Fig. 1G), suggesting that GLDC renders tumor cells resistant to CD8$^+$ T cell-mediated killing. These results suggest that GLDC-deficiency sensitizes tumor cells to CTL-mediated cytotoxicity.

Because GLDC is a key enzyme for glycine metabolism, we then investigated whether its enzymatic activity is required for the regulation of MHC-I expression. Human GLDC mutations of Gly763 to Asp (G763D, G768D in mouse) or Gly768 to Glu (G768E, G773E in mouse) have been reported to lose its glycine exchange activity (Bravo-Alonso et al, 2017). In GLDC-knockout A549 and H1299 cells, we expressed human GLDC variants (wild-type, G763D, or G768E). For mouse CT26- and MC38-EGFR$^{ex19del}$ cells, we introduced mouse GLDC variants (wild-type, G768D, or G773E). We found that reconstitution of human wild-type GLDC, GLDC$^{G763D}$ (or mouse GLDC$^{G768D}$) or GLDC$^{G768E}$ (or mouse GLDC$^{G773E}$) reduced MHC-I surface levels to similar levels (Fig. 1H; Appendix Fig. S2B–D). These results suggest that GLDC regulates MHC-I surface expression independent of its enzymatic activity.

## GLDC-deficiency promotes CD8$^+$ T cell-mediated immunosurveillance

We next investigated the biological functions of GLDC. Cell proliferation assays showed that GLDC-deficiency inhibited A549, H1299, CT26-EGFR$^{ex19del}$, or MC38-EGFR$^{ex19del}$ cell proliferation, which was reversed by reconstitution with human wild-type GLDC but not GLDC$^{G763D}$ (or mouse GLDC$^{G768D}$) or GLDC$^{G768E}$ (or mouse GLDC$^{G773E}$) (Appendix Fig. S3A). Consistently, tumor growth in immune-deficient NOD/ShiLtJGpt-Prkdc$^{em26Cd52}$Il2rg$^{em26Cd22}$/Gpt mice (NCG mice) subcutaneously injected with wild-type GLDC-reconstituted CT26-EGFR$^{ex19del}$ or MC38-EGFR$^{ex19del}$ cells was markedly faster than those injected with GLDC-deficient or GLDC$^{G773E}$-reconstituted cells (Fig. 2A,B). These results suggest that GLDC regulates tumor cell proliferation depending on its enzymatic activity.

As GLDC negatively regulates MHC-I antigen presentation independently of its enzymatic activity in in vitro experiments, it may function in cancer immunity. We subcutaneously injected CT26-EGFR$^{ex19del}$ or MC38-EGFR$^{ex19del}$ cells in immune-competent Balb/c or C57bl/6j mice, respectively, and showed that GLDC-deficiency in these cells suppressed tumor growth and prolonged animal survival, which was reversed by reconstitution with wild-type GLDC in these cells (Fig. 2C,D). In these experiments, we also found that reconstitution of GLDC-deficient cells with GLDC$^{G773E}$ promoted tumor growth to a lesser degree compared to reconstitution with wild-type GLDC (Fig. 2C,D). Depletion of CD8$^+$ T cells using anti-CD8 antibodies in Balb/c or C57bl/6j mice completely impaired the tumor-promoting effects induced by reconstitution of GLDC$^{G773E}$ in GLDC-deficient cells (Fig. 2E,F; Appendix Fig. S3B). These results suggest that antitumor CD8$^+$ T cell immunity contributes to tumor suppression induced by GLDC-deficiency. In addition, in line with the enhanced tumor burden in immune-competent Balb/c mice (Fig. 2G), in vivo analysis of tumors revealed that reconstitution of GLDC$^{G773E}$ in GLDC-deficient cells induced a decrease in MHC-I levels on the surface of tumor cells (Fig. 2H), numbers of CD8$^+$ T cells in TME

(Fig. 2I,J), and effector functions of tumor-infiltrating CD8$^+$ T cells in tumors, that was evidenced by their capacities to secrete IFNγ and granzyme B (GzmB) (Fig. 2K). Collectively, our data suggest GLDC inhibits MHC-I expression and CD8$^+$ T cell immunity independently of its enzymatic activity.

## GLDC suppresses STAT1-triggered MHC-I antigen presentation

We next investigated the mechanisms responsible for GLDC-mediated downregulation of MHC-I. Immunoblotting analysis showed that GLDC-deficiency upregulated the protein levels of HLA-A and B2M (Fig. 3A; Appendix Fig. S4A). Flow cytometry analysis showed that GLDC-deficiency did not affect the surface B2M internalization (Fig. 3B). Additionally, the protein synthesis inhibitor cycloheximide (CHX) treatment indicated that the half-life of HLA-A and B2M did not show marked changes after GLDC-deficiency (Fig. 3C), suggesting that GLDC does not affect their stability. Quantitative real-time PCR (qPCR) experiments indicated that GLDC-deficiency upregulated the mRNA levels of MHC-I genes, including *HLA-A*, *HLA-B*, *HLA-C,* and *B2M* (Fig. 3D, Appendix Fig. S4B). Meanwhile, the transcription of other major components involved in the MHC-I antigen presentation pathway and processing were also increased after GLDC-deficiency (Fig. 3D; Appendix Fig. S4B). Consistently, the mRNA levels of MHC-I genes were also enhanced in GLDC-deficient B16-OVA-EGFR$^{ex19del}$ cells in response to IFNγ (Fig. 3E). Together, these results suggest that GLDC regulates MHC-I levels at mRNA but not protein levels.

The transcription of major components involved in the antigen presentation pathway and processing are mainly regulated by the transcription factors STAT1 and NF-κB (Bell and Zou, 2024; Sykulev, 2023; Wang et al, 2024). It has been shown that activated STAT1 induces expression of the transcription factors IRF1 and NLRC5, which ultimately induce transcription of MHC-I genes such as *HLA-A*, *HLA-B*, *HLA-C* and *B2M* (Zhou, 2009). Reporter assays indicated that GLDC-deficiency induced GAS (IFNγ-activated site, STAT1 pathway) activation and had no marked effects on NF-κB activation (Fig. 3F; Appendix Fig. S4C). qPCR analysis indicated that GLDC-deficiency increased the mRNA levels of *IRF1* and *NLRC5* genes (Fig. 3G; Appendix Fig. S4D). The transcription of *Irf1* and *Nlrc5* genes were also enhanced in GLDC-deficient B16-OVA-EGFR$^{ex19del}$ cells in response to IFNγ (Fig. 3H). Knockout of STAT1 but not p65 impaired upregulation of MHC-I proteins such as HLA-A and B2M in GLDC-deficient A549 cells (Fig. 3I). Upregulation of the mRNA levels of *HLA-A*, *HLA-B*, *HLA-C*, *B2M*, *IRF1*, and *NLRC5* genes in GLDC-deficient cells was reversed by knockout of STAT1 (Fig. 3J). These results suggest that GLDC regulates transcription of MHC-I-related genes through the STAT1-IRF1/NLRC5 axis.

We next examined how GLDC regulates STAT1-triggered transcription of downstream genes. Phosphorylation is critical for the activation of STAT1; phosphorylated STAT1 forms homo-dimers, which translocate to the nucleus and bind to the conserved GASs on the target gene promoters to initiate their transcription (Villarino et al, 2017; Wei et al, 2017). Immunoblotting analysis showed that phosphorylation of STAT1$^{Y701}$ and STAT1$^{S727}$ were not markedly changed after GLDC-deficiency (Appendix Fig. S4E). IFNγ induced phosphorylation of STAT1$^{Y701}$ and STAT1$^{S727}$ in B16-OVA-EGFR$^{ex19del}$ cells, which was not affected by GLDC-deficiency

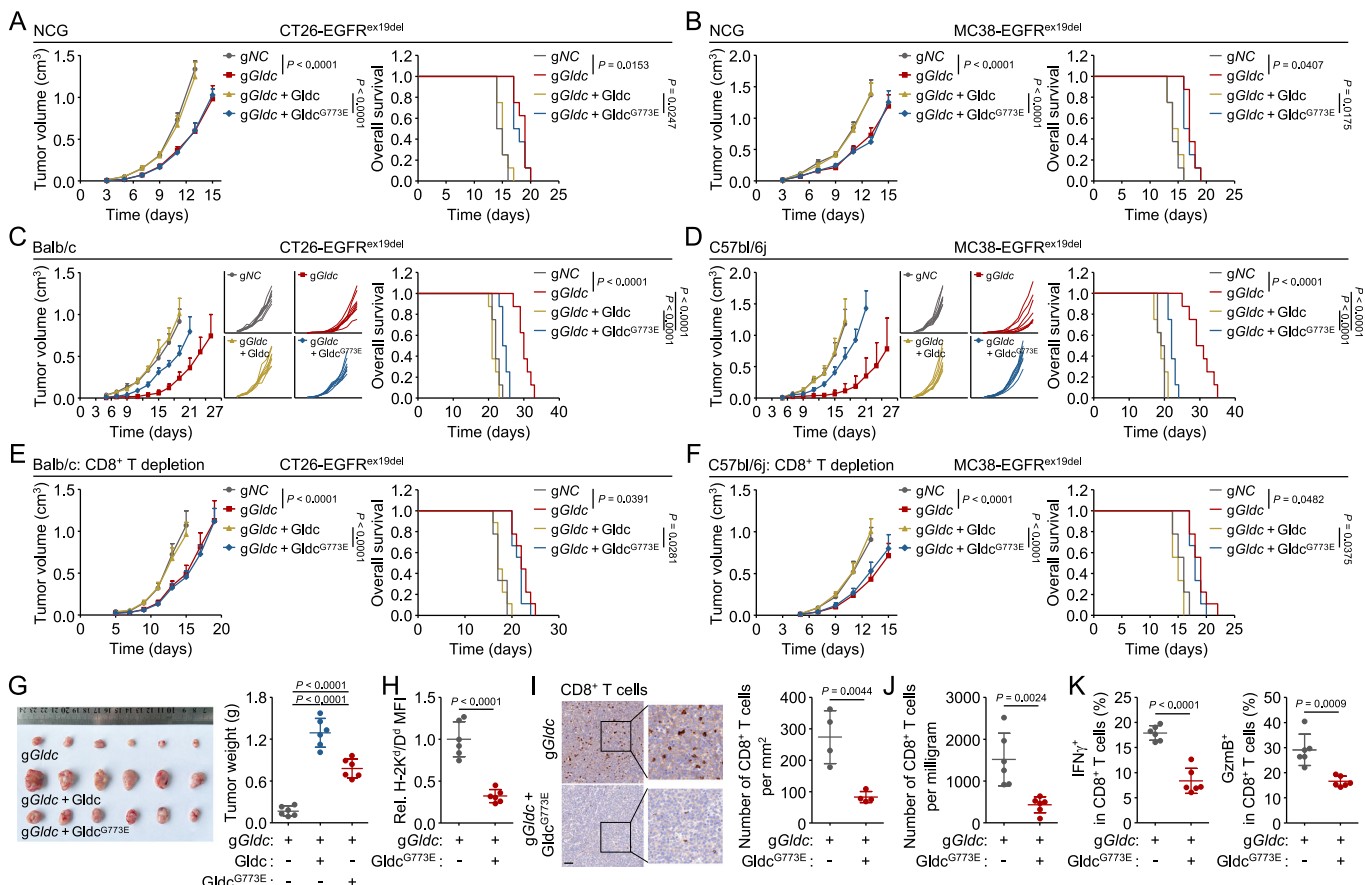

**Figure 2. GLDC-deficiency promotes CD8+ T cell-mediated immunosurveillance.**

(**A–F**) Effects of GLDC-deficiency on tumor growth. Control (gNC) or Gldc-deficient (gGldc) CT26-EGFR^ex19del (**A, C, E**) or MC38-EGFR^ex19del (**B, D, F**) cells (5 × 10^5) reconstituted with wild-type Gldc or Gldc^G773E were subcutaneously injected into mice. On day 3 after tumor cell inoculation, tumor sizes were measured every 2 days by caliper. Mice were sacrificed when the tumor size was bigger than 15 mm of the mean tumor diameter, tumor volume exceeded 2000 mm^3, or the tumor had ulcers with a diameter reached 10 mm. Graph shows mean ± SEM, $n = 8$ (**A–D**), $n = 9$ (**E, F**). Data were analyzed using two-way ANOVA with GraphPad Prism 8. Kaplan–Meier survival curves and corresponding log-rank (Mantel–Cox) tests were used to evaluate the statistical differences between groups in survival studies. There is a significant difference when the $P < 0.05$. (**G**) GLDC-deficiency suppresses tumor growth. The indicated CT26-EGFR^ex19del cells (5 × 10^5) were subcutaneously injected into Balb/c mice. Tumor-bearing mice were euthanized on day 18, and then the tumor tissues were separated from the mice. Tumor weights were measured by an analytical balance. Graph shows mean ± SEM, $n = 6$. Data were analyzed using a student's unpaired t-test with GraphPad Prism 8. (**H**) Effects of GLDC-deficiency on MHC-I surface expression of tumor cells in vivo. Gldc-deficient (gGldc) or Gldc^G773E-reconstituted CT26-EGFR^ex19del cells (5 × 10^5) were subcutaneously injected into Balb/c mice. Tumor-bearing mice were euthanized on day 20, and then the tumor tissues were separated from the mice. Tumor cells were isolated from the indicated tumor tissues, and then stained with the indicated antibodies and analyzed by flow cytometry. Graph shows mean ± SEM, $n = 6$ independent samples. Data were analyzed using a student's unpaired t-test with GraphPad Prism 8. (**I**) GLDC-deficiency increases the number of CD8+ T cells in tumor tissues. The indicated cells were subcutaneously injected into Balb/c mice. After 20 days, tumor-bearing mice were euthanized, and tumor tissues were analyzed. Representative images from IHC staining of CD8 in tumor sections were shown. Scale bar, 100 μm. Quantitative analysis of IHC data of CD8-positive cells were shown. Graph shows mean ± SEM, $n = 4$ independent samples. Data were analyzed using a student's unpaired t-test with GraphPad Prism 8. (**J, K**) Knockout of GLDC increases the number of CD8+ T cells in TME. The indicated cells were subcutaneously injected into Balb/c mice. Tumor-bearing mice were euthanized on day 20, and then the tumor tissues were separated from the mice. TILs were isolated from the indicated tumor tissues and stained with the indicated antibodies and analyzed by flow cytometry. CD8+ T cells were sorted from CD45+ populations. Graph shows mean ± SEM, $n = 6$ independent samples. Data were analyzed using a student's unpaired t-test with GraphPad Prism 8. Source data are available online for this figure.

(Appendix Fig. S4F). Cellular fractionation experiments showed that GLDC-deficiency also did not affect the nuclear translocation of STAT1 (Appendix Fig. S4G). Then, we investigated whether GLDC regulates the binding of STAT1 to promoter regions of downstream target genes. Chromatin immunoprecipitation (ChIP) experiments confirmed that STAT1 bound to the promoter region of the *IRF1* gene in A549 and H1299 cells, which was enhanced in GLDC-deficient cells (Fig. 3K; Appendix Fig. S4H). IFNγ induced the binding of STAT1 to the *IRF1* promoter region and it was enhanced in GLDC-deficient B16-OVA-EGFR^ex19del cells (Fig. 3L).

In addition, GLDC could not directly bind to the promoter region of *IRF1* gene either in control or STAT1-deficient cells (Appendix Fig. S4I). These results suggest that GLDC suppresses the binding of STAT1 to the promoter regions of downstream target genes.

## Phosphorylation and nuclear translocation of GLDC inhibit MHC-I expression

In our experiments, we found that GLDC had no marked effects on phosphorylation and nuclear translocation of STAT1, but inhibited the

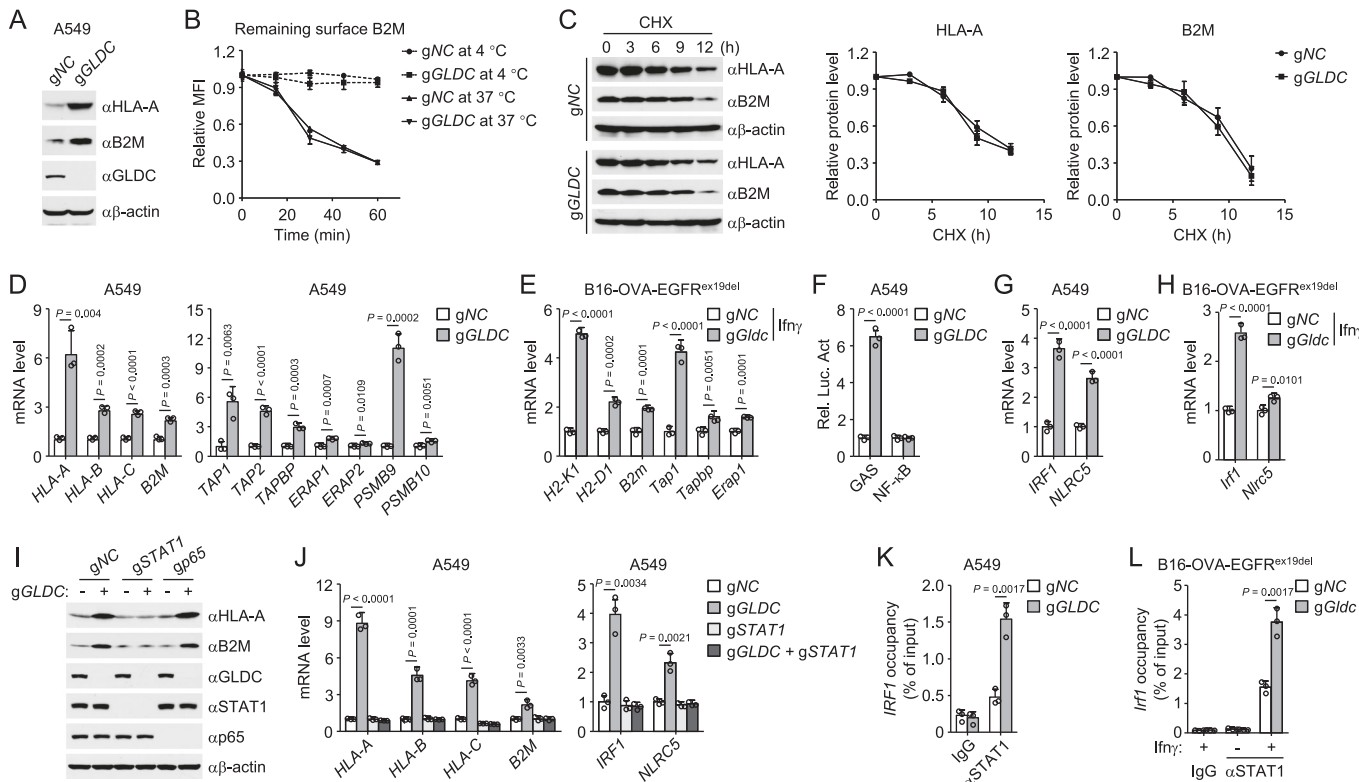

**Figure 3. GLDC suppresses STAT1-triggered MHC-I antigen presentation.**

(A) GLDC-deficiency upregulated the protein levels of HLA-A and B2M. The indicated A549 cells were cultured in the presence of EGF (20 ng/mL) for 24 h before immunoblotting analysis with the indicated antibodies. (B) Effects of GLDC-deficiency on surface B2M internalization. The indicated A549 cells were cultured in the presence of EGF (20 ng/mL) for 24 h before surface B2M internalization analysis. Graph shows mean ± SEM, $n = 3$ independent samples. Data were analyzed using two-way ANOVA with GraphPad Prism 8. (C) Effects of GLDC-deficiency on MHC-I degradation. The indicated A549 cells were cultured in the presence of EGF (20 ng/mL) for 24 h and then treated with CHX (0.1 mM) for the indicated times before immunoblotting analysis with the indicated antibodies. The HLA-A or B2M band intensities relative to the corresponding β-actin bands were shown in the histogram. Graph shows mean ± SEM, $n = 3$ independent samples. Data were analyzed using two-way ANOVA with GraphPad Prism 8. (D, E) Effects of GLDC-deficiency on the transcription of major components involved in the antigen presentation pathway and processing. The indicated A549 cells were cultured in the presence of EGF (20 ng/mL) for 24 h before qPCR analysis of mRNA levels of the indicated genes (D). The indicated B16-OVA-EGFR^ex19del cells were stimulated with Ifnγ (50 ng/mL) for 24 h before qPCR analysis of mRNA levels of the indicated genes (E). Graph shows mean ± SEM, $n = 3$ independent samples. Data were analyzed using a student's unpaired $t$-test with GraphPad Prism 8. (F) GLDC-deficiency activates GAS. The indicated A549 cells were cultured in the presence of EGF (20 ng/mL) and transfected with GAS or NF-kB reporter plasmids for 24 h before luciferase assays. Graph shows mean ± SEM, $n = 3$ independent samples. Data were analyzed using a student's unpaired t-test with GraphPad Prism 8. (G, H) Effects of GLDC-deficiency on transcription of *IRF1* and *NLRC5* genes. The indicated A549 cells were cultured in the presence of EGF (20 ng/mL) for 24 h before qPCR analysis of mRNA levels of the indicated genes (G). The indicated B16-OVA-EGFR^ex19del cells were stimulated with Ifnγ (50 ng/mL) for 24 h before qPCR analysis of mRNA levels of the indicated genes (H). Graph shows mean ± SEM, $n = 3$ independent samples. Data were analyzed using a student's unpaired $t$-test with GraphPad Prism 8. (I) Knockout of STAT1 inhibits upregulation of MHC-I induced by GLDC-deficiency. The indicated A549 cells were cultured in the presence of EGF (20 ng/mL) for 24 h before immunoblotting analysis with the indicated antibodies. (J) Knockout of STAT1 suppresses the transcription of MHC-I genes induced by GLDC-deficiency. The indicated A549 cells were cultured in the presence of EGF (20 ng/mL) for 24 h before qPCR analysis of mRNA levels of the indicated genes. Graph shows mean ± SEM, $n = 3$ independent samples. Data were analyzed using two-way ANOVA with GraphPad Prism 8. (K, L) Effects of GLDC-deficiency on the binding of STAT1 to *IRF1* promoter region. The indicated A549 cells were cultured in the presence of EGF (20 ng/mL) for 24 h before ChIP analysis. The indicated B16-OVA-EGFR^ex19del cells were stimulated with Ifnγ (50 ng/mL) for 24 h before ChIP analysis. The de-crosslinked DNA was subjected to qPCR analysis using specific primers. Graph shows mean ± SEM, $n = 3$ independent samples. Data were analyzed using two-way ANOVA with GraphPad Prism 8. Source data are available online for this figure.

binding of STAT1 to promoter regions of downstream target genes. However, GLDC is a key enzyme of the glycine cleavage system and has been reported to localize to mitochondria and cytoplasm (Liu et al, 2021). We hypothesized that GLDC is translocated to the nucleus and modulates the nuclear function of STAT1. The results in Fig. 1C–E prompt a potentiated role of GLDC in EGFR-mediated inhibition of MHC-I expression. Then, we serum-starved the cells and examined the subcellular location of GLDC following EGF stimulation. Confocal microscopy showed that EGF stimulation induced nuclear transloca-tion of GLDC (Fig. 4A). Cellular fractionation experiments further

confirmed that GLDC was accumulated in the nucleus after EGF treatment (Fig. 4B).

We next treated A549 cells with a panel of inhibitors, afatinib, U0126, Ly294002, ruxolitinib, saracatinib, and amuvatinib, which inhibited EGF-induced activation of EGFR, MEK, PI3K, JAK family kinases, SRC family kinases, and c-Kit, respectively. We found that Afatinib or Saracatinib treatment dramatically impaired the nuclear translocation of GLDC (Fig. 4C). Consistently, EGF stimulation induced tyrosine phosphorylation of GLDC (Fig. 4D; Appendix Fig. S5A,B). Afatinib or Saracatinib treatment abrogated EGF-induced phosphorylation of

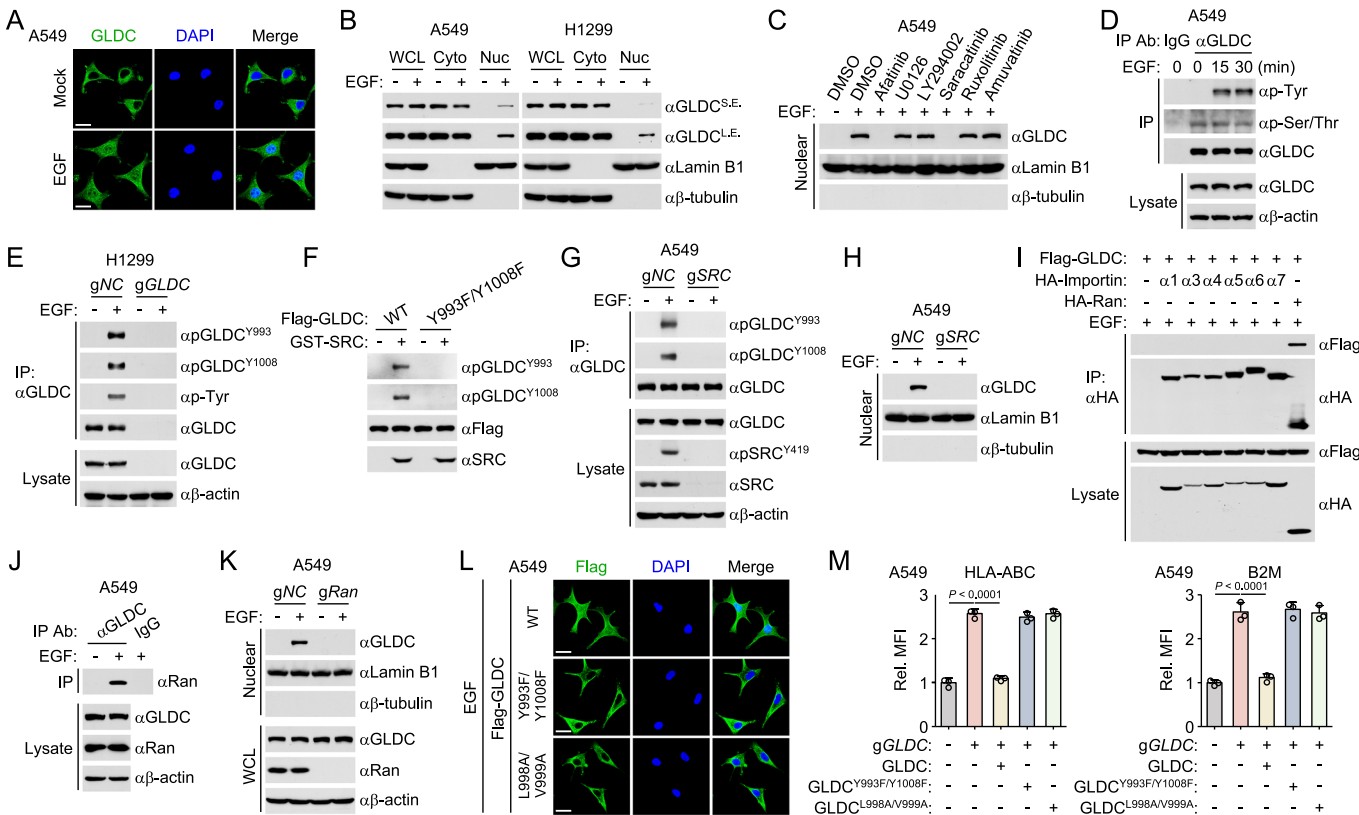

**Figure 4. Phosphorylation and nuclear translocation of GLDC inhibits MHC-I expression.**

(A) EGF stimulation induces GLDC nuclear translocation. A549 cells were serum-starved (12 h) and then treated with or without EGF (100 ng/ml) for 6 h. The cells were fixed with 4% paraformaldehyde and stained with anti-GLDC (green) and DAPI (blue). Images were obtained under objective of a confocal microscope. Representative images from IF staining are shown. Scale bar, 20 μm. (B) EGF treatment induces accumulation of GLDC in the nucleus. A549 or H1299 cells were serum-starved (12 h) and then treated with or without EGF (100 ng/ml) for 6 h. The cells were collected for subcellular fractionation experiments and immunoblotting analysis with the indicated antibodies. LE long exposure, SE short exposure. (C) Effects of a panel of inhibitors on EGF-induced GLDC nuclear translocation. A549 cells were serum-starved (12 h) and then treated with DMSO, Afatinib (2 μM), U0126 (10 μM), Ly294002 (20 μM), saracatinib (1 μM), ruxolitinib (5 μM), or amuvatinib (5 μM) for 2 h before EGF (100 ng/ml) treatment for 6 h. The cells were collected for subcellular fractionation experiments and immunoblotting analysis with the indicated antibodies. (D) EGF stimulation induces GLDC tyrosine phosphorylation. A549 cells were serum-starved (12 h) and then treated with or without EGF (100 ng/ml) for the indicated times before co-immunoprecipitation and immunoblotting analysis with the indicated antibodies. (E) GLDC^Y993 and GLDC^Y1008 are phosphorylated following EGF stimulation. The indicated H1299 cells were serum-starved (12 h) and then treated with or without EGF (100 ng/ml) for 30 min before co-immunoprecipitation and immunoblotting analysis with the indicated antibodies. (F) SRC mediates GLDC^Y993 and GLDC^Y1008 phosphorylation in vitro. The indicated recombinant proteins were incubated for in vitro kinase assays before immunoblotting analysis with the indicated antibodies. (G) Knockout of SRC inhibits EGF-induced phosphorylation of GLDC^Y993 and GLDC^Y1008. The indicated A549 cells were serum-starved (12 h) and then treated with or without EGF (100 ng/ml) for 30 min before co-immunoprecipitation and immunoblotting analysis with the indicated antibodies. (H) Knockout of SRC impairs EGF-induced GLDC nuclear translocation. The indicated A549 cells were serum-starved (12 h) and then treated with or without EGF (100 ng/ml) for 6 h before subcellular fractionation experiments and immunoblotting analysis with the indicated antibodies. (I) Ran is associated with GLDC. HEK293/EGFR cells were transfected with the indicated plasmids for 24 h and then treated with EGF (100 ng/ml) for 30 min before co-immunoprecipitation and immunoblotting analysis with the indicated antibodies. (J) EGF treatment induces the interaction of GLDC with Ran. A549 cells were serum-starved (12 h) and then treated with or without EGF (100 ng/ml) for 30 min before co-immunoprecipitation and immunoblotting analysis with the indicated antibodies. (K) Knockout of Ran impairs EGF-induced GLDC nuclear translocation. The indicated A549 cells were serum-starved (12 h) and then treated with or without EGF (100 ng/ml) for 6 h before subcellular fractionation experiments and immunoblotting analysis with the indicated antibodies. (L) Effects of GLDC mutations on its ability to translocate into the nucleus. A549 cells were cultured in the presence of EGF (50 ng/mL) and transfected with indicated plasmids for 20 h. The cells were fixed with 4% paraformaldehyde and stained with anti-Flag (green) and DAPI (blue). Images were obtained under objective of a confocal microscope. Representative images from IF staining are shown. Scale bar, 20 μm. (M) Effects of GLDC mutations on MHC-I surface expression. The indicated A549 cells were cultured in the presence of EGF (20 ng/mL) for 24 h before cytometry analysis with the indicated antibodies. Graph shows mean ± SEM, n = 3 independent samples. Data were analyzed using two-way ANOVA with GraphPad Prism 8. Source data are available online for this figure.

GLDC (Appendix Fig. S5C). We next attempted to identify the potential residues of GLDC that are phosphorylated following EGF treatment. We expressed Flag-tagged GLDC in HEK293T cells ectopically expressing EGFR (HEK293/EGFR), then GLDC protein was immunoprecipitated with anti-Flag antibody after EGF treatment and identified nine potentially phosphorylated tyrosine residues of GLDC by mass spectrometry (Appendix Table S2). Mutagenesis indicated that mutation of

Y993 or Y1008 but not the other seven tyrosine residues to phenylalanine impaired EGF-induced tyrosine phosphorylation of GLDC (Appendix Fig. S5D). Sequence analysis indicated that GLDC^Y993 and GLDC^Y1008 were conserved in various vertebrate species (Appendix Fig. S5E). To determine whether EGF mediates phosphorylation of GLDC^Y993 and GLDC^Y1008, we generated two rabbit polyclonal antibody specific for Y993-phosphorylated GLDC (pGLDC^Y993) and Y1008-

phosphorylated GLDC (pGLDC$^{Y1008}$), respectively. Immunoblotting analysis confirmed that GLDC$^{Y993}$ and GLDC$^{Y1008}$ were phosphorylated following EGF stimulation (Fig. 4E). EGF stimulation failed to induce GLDC$^{Y993F/Y1008F}$ phosphorylation (Appendix Fig. S5F). These results suggest that EGF mediates phosphorylation of GLDC at Y993 and Y1008. In addition, GLDC$^{Y993F/Y1008F}$ failed to be translocated into the nucleus upon EGF stimulation (Appendix Fig. S5G).

We next attempted to identify tyrosine kinases responsible for phosphorylation of GLDC$^{Y993}$ and GLDC$^{Y1008}$ upon EGFR activation. As shown in Appendix Fig. S5B,C, GLDC was not associated with EGFR, while inhibition of SRC family kinases abrogated EGF-induced phosphorylation of GLDC, suggesting that SRC family kinases are responsible for phosphorylation of GLDC. We co-transfected HEK293 cells with GLDC and different SRC family kinases. The results showed that overexpression of SRC but not the other SRC family kinases induced tyrosine phosphorylation of wild-type GLDC but not GLDC$^{Y993F/Y1008F}$ (Appendix Fig. S5H,I). In vitro phosphorylation assays further showed that SRC catalyzed tyrosine phosphorylation of wild-type GLDC, but not GLDC$^{Y993F/Y1008F}$ (Fig. 4F). Knockout of SRC impaired EGF-induced GLDC$^{Y993}$ and GLDC$^{Y1008}$ phosphorylation as well as its nuclear translocation (Fig. 4G,H). In addition, GLDC remained phosphorylated upon nuclear translocation (Appendix Fig. S5J). These results suggest that SRC catalyzes phosphorylation of GLDC at Y993 and Y1008, leading to its subsequent nuclear translocation.

Importin α, which has six family members (α1, α3, α4, α5, α6, and α7), is the best-characterized nuclear import pathways that binds NLS-containing proteins to facilitate their transport across the nuclear envelope (Mason et al, 2009). RanGDP-NTF2 complex is also a nuclear import system that mediates an importin-independent nuclear import pathway (Lu et al, 2014). Co-immunoprecipitation experiments indicated that Ran but not Importin αs was associated with GLDC (Fig. 4I). Endogenous co-immunoprecipitation further confirmed that EGF treatment induced the interaction of GLDC with Ran (Fig. 4J; Appendix Fig. S6A), while EGF treatment failed to induce the interaction of GLDC$^{Y993F/Y1008F}$ with Ran (Appendix Fig. S6B). Ran is a member of the Ras family of small GTPases, and exists intracellularly in both RanGDP and RanGTP forms. The asymmetric distribution of RanGDP and RanGTP between cytoplasm and nucleus determines that RanGDP and RanGTP mediate nuclear import and nuclear export, respectively (Lu et al, 2014). Ran$^{Q69L}$ is a dominant-negative mutant of Ran that is locked in a GTP-bound form and defective in nuclear import. EGF stimulation or overexpression of SRC induced the interaction of GLDC with wild-type Ran but not Ran$^{Q69L}$ (Appendix Fig. S6C,D), suggesting that GLDC is associated with RanGDP. In vitro GST pulldown assays further confirmed that wild-type GLDC but not GLDC$^{Y993F/Y1008F}$ was associated with RanGDP (Appendix Fig. S6E). Knockout of Ran abolished EGF-induced nuclear translocation of GLDC (Fig. 4K). Together, these results indicated that RanGDP mediates GLDC nuclear translocation.

To map the interaction residues, we expressed different GLDC truncation mutants and found that the aa994-1007 of GLDC was required for the interaction of GLDC with Ran induced by EGF (Appendix Fig. S6F). Mutations of hydrophobic residues in aa994-1007 of GLDC showed that GLDC$^{L998A/V999A}$ failed to interact with Ran (Appendix Fig. S6G), but retained the ability to be phosphorylated by SRC (Appendix Fig. S6H). These results suggest that L998/V999 of GLDC is important for its interaction with Ran. Confocal microscopy further indicated that GLDC$^{Y993F/Y1008F}$ or GLDC$^{L998A/V999A}$ failed to translocate into the nucleus following EGF treatment (Fig. 4L).

To determine whether phosphorylation and nuclear translocation of GLDC regulates MHC-I expression, we reconstituted GLDC-knockout A549, H1299, CT26-EGFR$^{ex19del}$ or MC38-EGFR$^{ex19del}$ cells with human wild-type GLDC, GLDC$^{Y993F/Y1008F}$ (or mouse GLDC$^{Y998F/Y1013F}$), or GLDC$^{L998A/V999A}$ (or mouse GLDC$^{L1003A/V1004A}$) (Appendix Fig. S6I). Flow cytometry analysis showed that reconstitution of wild-type GLDC but not GLDC$^{Y993F/Y1008F}$ (or mouse GLDC$^{Y998F/Y1013F}$) or GLDC$^{L998A/V999A}$ (or mouse GLDC$^{L1003A/V1004A}$) abrogated the upregulation of MHC-I in GLDC-deficient cells (Fig. 4M; Appendix Fig. S6J). Taken together, these results suggest that phosphorylation and nuclear translocation of GLDC inhibits MHC-I surface expression.

## Nuclear GLDC inhibits MHC-I expression by hijacking SMARCE1

To investigated how nuclear GLDC regulates MHC-I expression, we expressed Flag-tagged GLDC in HEK293/EGFR cells, and then GLDC-bound proteins were immunoprecipitated with anti-Flag antibody and analyzed by mass spectrometry. In comparison to mock, 72 proteins were identified as GLDC high-confidence interactors after EGF treatment (Appendix Table S3). Among these proteins, SMARCE1 was found to have a strong interaction with GLDC after EGF stimulation (Appendix Table S3). EGF treatment or overexpression of SRC induced the interaction of SMARCE1 with wild-type GLDC but not GLDC$^{Y993F/Y1008F}$ or GLDC$^{L998A/V999A}$ (Appendix Fig. S7A,B). Endogenous co-immunoprecipitation experiments further indicated that EGF induced the interaction of GLDC with SMARCE1 in the nucleus (Fig. 5A). SMARCE1 is part of the large ATP-dependent chromatin remodeling complex SWI/SNF and involved in transcriptional activation and repression of select genes by chromatin remodeling (Mashtalir et al, 2018). Co-immunoprecipitation experiments confirmed that SMARCE1 interacted with the core components of the SWI/SNF complex, and EGF stimulation had no marked effects on these associations (Appendix Fig. S7C). To determine if GLDC is associated with other components of the SWI/SNF complex, we performed co-immunoprecipitation experiments between GLDC and the core components of the SWI/SNF complex. The results showed that GLDC was only associated with SMARCE1 but not the other core components of the SWI/SNF complex, including SMARCB1, SMARCD1, SMARCA4, BRD7, BRD9 and ACTL6A (Appendix Fig. S7D). These results suggest that GLDC is associated with SMARCE1 but not the other components of the SWI/SNF complex.

We next investigated the function of SMARCE1 in GLDC-regulated MHC-I expression. SMARCE1-deficiency induced upregulation of the binding of STAT1 to *IRF1* promoter region, the mRNA levels of *IRF1* and *NLRC5* genes, and the surface expression of MHC-I to a lesser degree compared to GLDC-deficiency (Fig. 5B–D; Appendix Fig. S7E). In these experiments, we also found that knockout of GLDC failed to upregulate MHC-I in SMARCE1-deficient cells (Fig. 5B–D), suggesting that GLDC inhibits MHC-I expression through SMARCE1. We then explored the residues of SMARCE1 required for its interaction with GLDC. We expressed different SMARCE1 truncation mutants and found that the aa67-134 of SMARCE1 was required for the interaction of SMARCE1 with GLDC induced by EGF (Appendix Fig. S7F). Mutations of hydrophobic residues in aa66-134 of SMARCE1 showed that SMARCE1$^{L93A/I96A/I99A}$ failed to interact with GLDC after EGF treatment (Appendix Fig. S7G, H), suggesting that L93/I96/I99 of SMARCE1 is important for its interaction with GLDC. SMARCE1 contains a N-terminal high-mobility group (HMG) DNA-

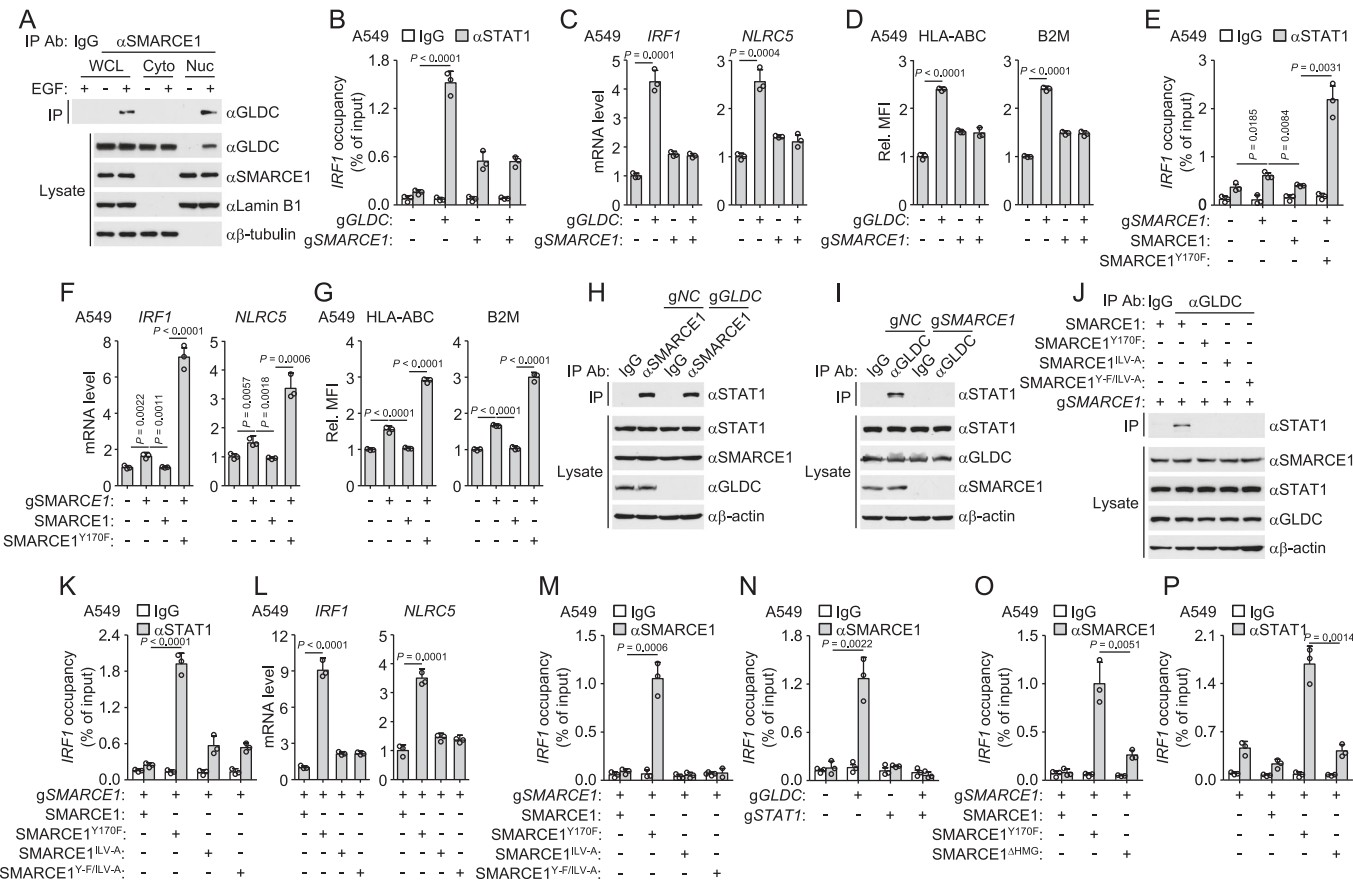

**Figure 5. Nuclear GLDC inhibits MHC-I expression by hijacking SMARCE1.**

(A) EGF induces the interaction of GLDC with SMARCE1 in nucleus. A549 cells were serum-starved (12 h) and then treated with or without EGF (100 ng/ml) for 6 h before subcellular fractionation experiments. Subcellular fractions were co-immunoprecipitated and analyzed by immunoblotting with the indicated antibodies. (B–D) Effects of SMARCE1-deficiency on the binding of STAT1 to *IRF1* promoter region, the transcription of *IRF1* and *NLRC5* genes or the surface expression of MHC-I. The indicated A549 cells were cultured in the presence of EGF (20 ng/mL) for 24 h before ChIP analysis (B), qPCR analysis of mRNA levels of the indicated genes (C), or flow cytometry analysis with the indicated antibodies (D). Graph shows mean ± SEM, $n = 3$ independent samples. Data were analyzed using two-way ANOVA with GraphPad Prism 8. (E–G) Effects of SMARCE1 mutations on the binding of STAT1 to *IRF1* promoter region, the transcription of *IRF1* and *NLRC5* genes or the surface expression of MHC-I. The indicated A549 cells were cultured in the presence of EGF (20 ng/mL) for 24 h before ChIP analysis (E), qPCR analysis of mRNA levels of the indicated genes (F), or flow cytometry analysis with the indicated antibodies (G). Graph shows mean ± SEM, $n = 3$ independent samples. Data were analyzed using two-way ANOVA with GraphPad Prism 8. (H) SMARCE1 is associated with STAT1. The indicated A549 cells were cultured in the presence of EGF (20 ng/mL) for 24 h before co-immunoprecipitation and immunoblotting analysis with the indicated antibodies. (I) Effects of SMARCE1-deficiency on the interaction of GLDC with STAT1. The indicated A549 cells were cultured in the presence of EGF (20 ng/mL) for 24 h before co-immunoprecipitation and immunoblotting analysis with the indicated antibodies. (J–L) Effects of SMARCE1 mutations on the interaction of GLDC with STAT1, the binding of STAT1 to *IRF1* promoter region or the transcription of the *IRF1* and *NLRC5* genes. The indicated cells were cultured in the presence of EGF (20 ng/mL) for 24 h before co-immunoprecipitation and immunoblotting analysis with the indicated antibodies (J), ChIP analysis and qPCR analysis using specific primers (K), or PCR analysis of mRNA levels of the indicated genes (L). Graph shows mean ± SEM, $n = 3$ independent samples. Data were analyzed using two-way ANOVA with GraphPad Prism 8. (M–O) Effects of SMARCE1 mutations, GLDC-deficiency or STAT1-deficiency on the binding of SMARCE1 to the *IRF1* promoter region. The indicated cells were cultured in the presence of EGF (20 ng/mL) for 24 h before ChIP analysis. The de-crosslinked DNA was subjected to qPCR analysis using specific primers. Graph shows mean ± SEM, $n = 3$ independent samples. Data were analyzed using two-way ANOVA with GraphPad Prism 8. (P) SMARCE1$^{\Delta HMG}$ impairs the binding of STAT1 to the *IRF1* promoter region. The indicated cells were cultured in the presence of EGF (20 ng/mL) for 24 h before ChIP analysis. The de-crosslinked DNA was subjected to qPCR analysis using specific primers. Graph shows mean ± SEM, $n = 3$ independent samples. Data were analyzed using two-way ANOVA with GraphPad Prism 8. Source data are available online for this figure.

binding domain (aa66-134), which contains three α helix structures (A: aa72-87, B: aa93-105, C: aa109-127) (St Pierre et al, 2022). The α helix structures of the HMG domain determine its DNA-binding activity (Stros et al, 2007). Our results suggest that GLDC is associated with the B α helix structure of the HMG domain of SMARCE1.

In our experiments, we also found that EGF stimulation increased SMARCE1 protein levels, which was downregulated after Afatinib or Saracatinib treatment (Appendix Fig. S8A,B). Immunoblotting analysis indicated that EGF stimulation induced tyrosine

phosphorylation of SMARCE1 (Appendix Fig. S8C). Knockout of SRC abrogated EGF-induced phosphorylation of SMARCE1 (Appendix Fig. S8D). Analysis of proteomic databases indicates that 8 tyrosine residues of SMARCE1 are potentially phosphorylated (https://www.phosphosite.org/). Mutagenesis indicated that mutations of Y73 and Y170, but not the other six tyrosine residues, to phenylalanine impaired EGF-induced tyrosine phosphorylation of SMARCE1 (Appendix Fig. S8E). Sequence analysis indicated that SMARCE1$^{Y73}$ and SMARCE1$^{Y170}$ were conserved in various

vertebrate species (Appendix Fig. S8F). SMARCE1$^{Y73F/170F}$ failed to be phosphorylated by EGF stimulation or overexpression of SRC (Appendix Fig. S8G,H). SMARCE1$^{Y170F}$ but not SMARCE1$^{Y73F}$ failed to interact with GLDC (Appendix Fig. S8I,J), suggesting that SRC-mediated phosphorylation of SMARCE1$^{Y170}$ is required for the interaction of SMARCE1 with GLDC. SMARCE1$^{Y73F}$ but not SMARCE1$^{Y170F}$ had a shorter half-life in CHX-treated A549 cells (Appendix Fig. S8K). SMARCE1$^{Y73F}$ but not SMARCE1$^{Y170F}$ failed to be upregulated following EGF treatment (Appendix Fig. S8L). These results suggest that phosphorylation of SMARCE1$^{Y73}$ inhibits its degradation. We generated two rabbit polyclonal antibodies specific for Y73-phosphorylated SMARCE1 (pSMARCE1$^{Y73}$) and Y170-phosphorylated SMARCE1 (pSMARCE1$^{Y170}$), respectively. Immunoblotting analysis confirmed that SMARCE1$^{Y73}$ and SMARCE1$^{Y170}$ were phosphorylated after overexpression of SRC (Appendix Fig. S8M). Together, these results indicate that SRC catalyzes phosphorylation of SMARCE1 at Y73 and Y170, SMARCE1$^{Y73}$ and SMARCE1$^{Y170}$ phosphorylation enhances its stability and induces its interaction with GLDC, respectively.

Next, we reconstituted SMARCE1-knockout A549 cells with wild-type SMARCE1 or SMARCE1$^{Y170F}$ and found that SMARCE1-deficiency caused upregulation of the binding of STAT1 to *IRF1* promoter region, the mRNA levels of *IRF1* and *NLRC5* genes and the surface expression of MHC-I were abrogated by reconstitution with wild-type SMARCE1, but dramatically enhanced in SMARCE1$^{Y170F}$-reconstituted cells (Fig. 5E–G; Appendix Fig. S8N). These results suggest that SMARCE1 promotes STAT1-triggered transcription of downstream genes, while is inhibited by the interaction of GLDC with SMARCE1 in the nucleus.

We further investigated how SMARCE1 regulates the binding of STAT1 to promoter regions of downstream target genes. Previous studies have demonstrated that SMARCE1 can regulate gene transcription by binding directly to transcription factors (Sokol et al, 2017). Endogenous co-immunoprecipitation experiments showed that SMARCE1 was associated with STAT1, which was not affected by GLDC-deficiency (Fig. 5H; Appendix Fig. S9A). GLDC was also associated with STAT1, but it was not observed in SMARCE1-deficient cells (Fig. 5I; Appendix Fig. S9B). In addition, mutation of SMARCE1$^{Y73/170}$ to phenylalanine or overexpression of SRC did not affect the interaction of SMARCE1 with STAT1 (Appendix Fig. S9C). Domain mapping experiments indicated that aa174-220 of SMARCE1 was required for the interaction of SMARCE1 with STAT1 (Appendix Fig. S9D). Mutations of hydrophobic residues in aa174-220 of SMARCE1 showed that SMARCE1$^{I206A/L207A/V218A}$ (SMARCE1$^{ILV-A}$) failed to interact with STAT1 (Appendix Fig. S9E,F), suggesting that I206/L207/V218 of SMARCE1 is important for its interaction with STAT1. We then reconstituted SMARCE1-knockout A549 cells with wild-type SMARCE1, SMARCE1$^{Y170F}$, SMARCE1$^{ILV-A}$ or SMARCE1$^{Y170F}$ $^{&}$ $^{I206A/L207A/V218A}$ (SMARCE1$^{Y-F/ILV-A}$). Co-immunoprecipitation experiments showed that wild-type SMARCE1 was associated with GLDC and STAT1 (Appendix Fig. S9G). SMARCE1$^{ILV-A}$ was only associated with GLDC, and abrogated the interaction of STAT1 with GLDC (Fig. 5J; Appendix Fig. S9G), suggesting that the interaction between SMARCE1 and STAT1 is necessary for STAT1 interacting with GLDC. In addition, the binding of STAT1 to the *IRF1* promoter region and the mRNA levels of *IRF1* and *NLRC5* genes in SMARCE1$^{Y-F/ILV-A}$-reconstituted cells were lower than that of SMARCE1$^{Y170F}$-reconstituted cells, but

similar to SMARCE1$^{ILV-A}$-reconstituted cells (Fig. 5K,L). Together, these results suggest that SMARCE1 enhances the binding of STAT1 to promoter regions of downstream target genes by interacting with STAT1.

Furthermore, SMARCE1 bound to the promoter region of the *IRF1* gene in SMARCE1$^{Y170F}$-reconstituted cells but not in wild-type SMARCE1, SMARCE1$^{ILV-A}$ or SMARCE1$^{Y-F/ILV-A}$-reconstituted cells (Fig. 5M). GLDC-deficiency induced the binding of SMARCE1 to the *IRF1* promoter region, but it was not observed in STAT1-deficient cells (Fig. 5N). These results suggest that SMARCE1 binds to the promoter region of the *IRF1* gene through STAT1, and which is inhibited by the interaction of GLDC with SMARCE1 in the nucleus. We further explored whether the DNA-binding activity of SMARCE1 is required for its function to enhance the binding of STAT1 to the *IRF1* promoter region. We reconstituted SMARCE1-knockout A549 cells with wild-type SMARCE1, SMARCE1$^{Y170F}$ or SMARCE1$^{ΔHMG}$ and found that SMARCE1$^{ΔHMG}$ was associated with STAT1 but not GLDC (Appendix Fig. S9H). The binding of SMARCE1$^{ΔHMG}$ to the *IRF1* promoter region was significantly weaker than SMARCE1$^{Y170F}$ (Fig. 5O). The binding of STAT1 to the promoter region of the *IRF1* gene in SMARCE1$^{ΔHMG}$-reconstituted cells was significantly lower than that in SMARCE1$^{Y170F}$-reconstituted cells, and similar to SMARCE1-deficient cells (Fig. 5P), suggesting that SMARCE1-mediated enhancement of STAT1 binding to promoter regions of downstream target genes is dependent on its DNA-binding activity. Taken together, SMARCE1 is a co-activator of STAT1, and the nuclear GLDC inhibits STAT1-triggered MHC-I expression by hijacking SMARCE1.

## Nuclear GLDC induces promoter DNA hypermethylation of STAT1-targeted genes by recruiting DNMT1/DMAP1

Among candidate proteins for mass spectrometry, we also noticed that DMAP1 had a strong interaction with GLDC following EGF treatment (Appendix Table S3). DMAP1 is a DNMT1-associated protein (Rountree et al, 2000). DNMT1 is an essential DNA methyltransferase that catalyzes the transfer of methyl groups to CpG islands in DNA and mediates transcriptional repression. DMAP1 can interact with DNMT1 to form a repressive transcription complex (Rountree et al, 2000). EGF treatment or overexpression of SRC induced the interaction of DMAP1 with wild-type GLDC but not GLDC$^{Y993F/Y1008F}$ or GLDC$^{L998A/V999A}$ (Appendix Fig. S10A,B). Endogenous co-immunoprecipitation experiments further indicated that DMAP1 was associated with GLDC in the nucleus after EGF treatment (Appendix Fig. S10C). The promoter regions of the *IRF1* and *NLRC5* genes contain a large number of CpG islands. Analysis of methylation status of promoter regions of *IRF1* and *NLRC5* genes by MeDIP showed that DNA hypermethylation (5mC) was concentrated in promoter regions of *IRF1* and *NLRC5* genes, which was downregulated by knockout of GLDC (Fig. 6A). Decitabine (DAC, DNMTs inhibitor) treatment removed the DNA methylation of *IRF1* promoter region (Fig. 6B). IFNγ treatment increased the levels of *IRF1* promoter DNA methylation, which were observed in GLDC-deficient cells (Fig. 6B). In this experiment, we also observed that the levels of *IRF1* promoter DNA methylation in DAC-treated A549 cells were not downregulated by knockout of GLDC (Fig. 6B). Furthermore, knockout of DMAP1 or DNMT1 in A549 cells downregulated the levels of *IRF1* promoter DNA methylation, which was not further downregulated by GLDC-

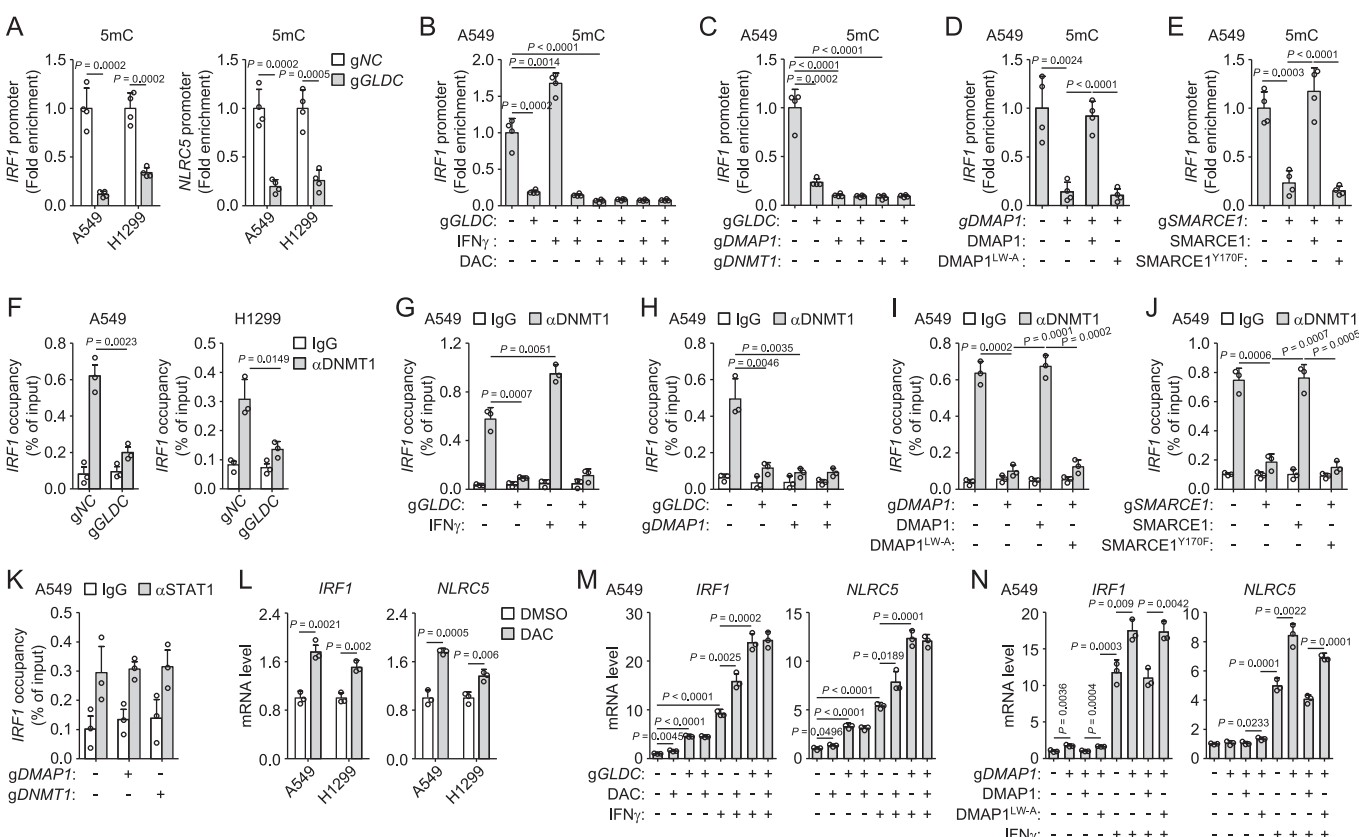

**Figure 6. Nuclear GLDC induces promoter DNA hypermethylation of STAT1-targeted genes by recruiting DNMT1/DMAP1.**

(A) Effects of GLDC-deficiency on the methylation status of *IRF1* and *NLRC5* promoter regions. The indicated A549 or H1299 cells were cultured in the presence of EGF (20 ng/mL) for 24 h before MeDIP analysis. The de-crosslinked DNA was subjected to qPCR analysis using specific primers. Graph shows mean ± SEM, n = 4 independent samples. Data were analyzed using a student's unpaired *t*-test with GraphPad Prism 8. (B) Effects of DAC treatment on the methylation status of the *IRF1* promoter region. The indicated A549 cells were treated with DMSO or DAC (100 nM) in the presence of EGF (20 ng/mL) for 7 days and then stimulated with IFNγ (50 ng/mL) for 24 h before MeDIP analysis. The de-crosslinked DNA was subjected to qPCR analysis using specific primers. Graph shows mean ± SEM, n = 4 independent samples. Data were analyzed using two-way ANOVA with GraphPad Prism 8. (C–E) Effects of DMAP1, DNMT1, or SMARCE1 on the methylation status of the *IRF1* promoter region. The indicated A549 cells were cultured in the presence of EGF (20 ng/mL) for 24 h before MeDIP analysis. The de-crosslinked DNA was subjected to qPCR analysis using specific primers. Graph shows mean ± SEM, n = 4 independent samples. Data were analyzed using two-way ANOVA with GraphPad Prism 8. (F) DNMT1 binds to the promoter region of the *IRF1* gene. The indicated A549 or H1299 cells were cultured in the presence of EGF (20 ng/mL) for 24 h before ChIP analysis. The de-crosslinked DNA was subjected to qPCR analysis using specific primers. Graph shows mean ± SEM, n = 3 independent samples. Data were analyzed using two-way ANOVA with GraphPad Prism 8. (G) IFNγ treatment enhances the binding of DNMT1 to the *IRF1* promoter region. The indicated A549 cells were cultured in the presence of EGF (20 ng/mL) for 24 h and then stimulated with IFNγ (50 ng/mL) for 24 h before ChIP analysis. The de-crosslinked DNA was subjected to qPCR analysis using specific primers. Graph shows mean ± SEM, n = 3 independent samples. Data were analyzed using two-way ANOVA with GraphPad Prism 8. (H–J) Effects of DMAP1 or SMARCE on the binding of DNMT1 to the *IRF1* promoter region. The indicated A549 cells were cultured in the presence of EGF (20 ng/mL) for 24 h before ChIP analysis. The de-crosslinked DNA was subjected to qPCR analysis using specific primers. Graph shows mean ± SEM, n = 3 independent samples. Data were analyzed using two-way ANOVA with GraphPad Prism 8. (K) Effects of DMAP1 or DNMT1-deficiency on the binding of STAT1 to the *IRF1* promoter region. The indicated A549 cells were cultured in the presence of EGF (20 ng/mL) for 24 h before ChIP analysis. The de-crosslinked DNA was subjected to qPCR analysis using specific primers. Graph shows mean ± SEM, n = 3 independent samples. Data were analyzed using two-way ANOVA with GraphPad Prism 8. (L) DAC treatment up-regulates the mRNA levels of *IRF1* and *NLRC5* genes. A549 or H1299 cells were treated with DMSO or DAC (100 nM) in the presence of EGF (20 ng/mL) for 7 days before qPCR analysis of mRNA levels of the indicated genes. Graph shows mean ± SEM, n = 3 independent samples. Data were analyzed using two-way ANOVA with GraphPad Prism 8. (M) DAC treatment enhances IFNγ-induced transcription of *IRF1* and *NLRC5* genes. The indicated A549 cells were treated with DMSO or DAC (100 nM) in the presence of EGF (20 ng/mL) for 7 days and then stimulated with IFNγ (50 ng/mL) for 24 h before qPCR analysis of mRNA levels of the indicated genes. Graph shows mean ± SEM, n = 3 independent samples. Data were analyzed using two-way ANOVA with GraphPad Prism 8. (N) Effects of DMAP1 mutations on the transcription of *IRF1* and *NLRC5* genes. The indicated A549 cells were cultured in the presence of EGF (20 ng/mL) for 24 h and then stimulated with IFNγ (50 ng/mL) for 24 h before qPCR analysis of mRNA levels of the indicated genes. Graph shows mean ± SEM, n = 3 independent samples. Data were analyzed using two-way ANOVA with GraphPad Prism 8. Source data are available online for this figure.

deficiency (Fig. 6C; Appendix Fig. S10D). Together, these results suggest that GLDC mediates *IRF1* promoter DNA methylation through DMAP1/DNMT1. We then explored the residues of DMAP1 required for its association with GLDC. Domain mapping experiments showed that aa141-211 of DMAP1 was required for the

interaction of DMAP1 with GLDC induced by overexpression of SRC (Appendix Fig. S10E). Mutations of hydrophobic residues in aa141-211 of DMAP1 showed that DMAP1[L145A/W152A] failed to interact with GLDC after overexpression of SRC (Appendix Fig. S10F,G), suggesting that L145/W152A of DMAP1 is important for

its interaction with GLDC. Next, we reconstituted DMAP1-knockout A549 cells with wild-type DMAP1 or DMAP1$^{L145A/W152A}$ (DMAP1$^{LW-A}$) and showed that the downregulation of *IRF1* promoter DNA methylation in DMAP1-deficient cells was reversed in the cells reconstituted by wild-type DMAP1 but not DMAP1$^{LW-A}$ (Fig. 6D; Appendix Fig. S10H). These results suggest that GLDC mediates *IRF1* promoter DNA methylation through interaction with DMAP1. In addition, SMARCE1-deficiency downregulated the levels of *IRF1* promoter DNA methylation, which was reversed in the cells reconstituted by wild-type SMARCE1 but not SMARCE1$^{Y170F}$ (Fig. 6E), suggesting that the interaction between GLDC and SMRACE1 is also required for GLDC-induced promoter DNA methylation of STAT1-targeted genes.

Next, we tested whether DNMT1 binds to the promoter regions of STAT1-triggered genes. ChIP experiments indicated that DNMT1 bound to the promoter region of the *IRF1* gene, which was inhibited by GLDC-deficiency (Fig. 6F). IFNγ treatment enhanced the binding of DNMT1 to the *IRF1* promoter region, but it was not observed in GLDC-deficient cells (Fig. 6G). Knockout of DMAP1 inhibited the binding of DNMT1 to the *IRF1* promoter region, which was not further downregulated by GLDC-deficiency (Fig. 6H; Appendix Fig. S10I). The downregulation of the binding of DNMT1 to the *IRF1* promoter region in DMAP1-deficient cells was reversed in the cells reconstituted by wild-type DMAP1 but not DMAP1$^{L145A/W152A}$ (DMAP1$^{LW-A}$) (Fig. 6I). These results suggest that the interaction of GLDC with DMAP1 recruits DNMT1 to the *IRF1* promoter region. Consistently, SMARCE1-deficiency downregulated the binding of DNMT1 to the *IRF1* promoter region, which was reversed in the cells reconstituted by wild-type SMARCE1 but not SMARCE1$^{Y170F}$ (Fig. 6J). Next, we investigated whether DMAP1/DNMT1-induced DNA hypermethylation affects the binding of STAT1 to the promoter regions of downstream target genes. CHIP experiments showed that the binding of STAT1 to the *IRF1* promoter region in DMAP1 or DNMT1 knockout cells did not show a significant difference in comparison to control cells (Fig. 6K), suggesting that the DMAP1/DNMT1-induced promoter DNA hypermethylation does not affect the binding of STAT1 to promoter regions of downstream target genes. In addition, we also investigated whether DMAP1/DNMT1-induced DNA hypermethylation affects the function of SMARCE1 as a co-activator of STAT1. CHIP experiments showed that binding of SMARCE1 to the *IRF1* promoter region in DMAP1 knockout cells did not show a significant difference in comparison to control cells, and knockout of DMAP1 did not affect the binding of STAT1 or SMARCE1 to the *IRF1* promoter region in GLDC-deficient cells (Appendix Fig. S10J,K). DMAP1-deficiency did not affect the interaction of GLDC with SMARCE1, and the interaction between GLDC and DMAP1 was also not affected by SMARCE1-deficiency (Appendix Fig. S10L,M). These suggest that the function of SMARCE1 as a co-activator of STAT1 is not affected by DMAP1.

We next assessed the functional role of DMAP1/DNMT1 in GLDC-regulated MHC-I expression. qPCR analysis indicated that DAC treatment upregulated mRNA levels of *IRF1* and *NLRC5* genes, and potentiated IFNγ-induced transcription of *IRF1* and *NLRC5* genes (Fig. 6L,M). However, GLDC-deficiency impaired the upregulation of mRNA levels of *IRF1* and *NLRC5* genes induced by DAC treatment (Fig. 6M). In these experiments, we also observed that the mRNA levels of *IRF1* and *NLRC5* genes in DAC-treated

cells were lower than in GLDC-deficient cells (Fig. 6M). These results suggest that DMAP1/DNMT1-mediated transcriptional repression of *IRF1* and *NLRC5* genes is part of the inhibition function induced by GLDC. Consistently, DMAP1-deficiency increased mRNA levels of *IRF1* and *NLRC5* genes, and potentiated IFNγ-induced transcription of *IRF1* and *NLRC5* genes (Fig. 6N). Upregulation of mRNA levels of *IRF1* and *NLRC5* genes in DMAP1-deficient cells were abrogated in the cells reconstituted by wild-type DMAP1 but not DMAP1$^{LW-A}$ (Fig. 6N). Taken together, the nuclear GLDC/SMARCE1/STAT1 complex recruits DNMT1/DMAP1 to the promoter regions of *IRF1* and *NLRC5* genes, resulting in hypermethylation of their promoters and transcriptional inhibition of the downstream genes.

## GLDC-deficiency sensitizes the antitumor effects of PD-1 blockade therapy

We next investigated the biological functions of GLDC phosphorylation and nuclear translocation. Cell proliferation assays showed that reconstitution of GLDC-deficient CT26-EGFR$^{ex19del}$ or MC38-EGFR$^{ex19del}$ cells with mouse GLDC$^{Y998F/Y1013F}$ (GLDC$^{Y-F}$) or GLDC$^{L1003A/V1004A}$ promoted cell proliferation to a similar degree compared to reconstitution with wild-type GLDC (Appendix Fig. S11). Consistently, tumor growth and animal survival in NCG mice subcutaneously injected with GLDC$^{Y-F}$-reconstituted CT26-EGFR$^{ex19del}$ or MC38-EGFR$^{ex19del}$ cells did not show a significant difference in comparison to those injected with wild-type GLDC-reconstituted cells (Fig. 7A). These results suggest that phosphorylation and nuclear translocation of GLDC did not affect tumor cell proliferation. Next, we subcutaneously injected the wild-type GLDC or GLDC$^{Y-F}$-reconstituted CT26-EGFR$^{ex19del}$ or MC38-EGFR$^{ex19del}$ cells in immune-competent Balb/c or C57bl/6j mice respectively, and found that reconstitution of GLDC$^{Y-F}$ in these cells suppressed tumor growth and improved the overall survival compared to reconstitution of wild-type GLDC in these cells (Fig. 7B). Depletion of CD8$^+$ T cells completely abrogated the tumor-suppressive function induced by reconstitution of GLDC$^{Y-F}$ in GLDC-deficient cells compared to reconstitution of wild-type GLDC (Fig. 7C). These results suggest that GLDC-mediated CD8$^+$ T cells immunosuppressive function is dependent on phosphorylation of GLDC$^{Y993/Y1008}$. In addition, in line with the reduced tumor burden (Fig. 7D), in vivo analysis of GLDC$^{Y-F}$-reconstituted tumors revealed higher MHC-I levels on tumor cells compared to wild-type GLDC-reconstituted tumors (Fig. 7E), accompanied by increased numbers of CD8$^+$ T cells in the TME (Fig. 7F–H). The tumor-infiltrating CD8$^+$ T cells in GLDC$^{Y-F}$-reconstituted tumors had increased effector functions compared to those from the parental wild-type GLDC-reconstituted tumors, evidenced by their capacities to secrete IFNγ and GzmB (Fig. 7I). Collectively, our data suggest that phosphorylation of GLDC$^{Y993/Y1008}$ inhibits MHC-I expression and CD8$^+$ T cell immunity.

In patients treated with immunotherapy, antigen expression and presentation levels are known to be critical factors of treatment response and an issue addressed in multiple response biomarker studies (Sykulev, 2023). Considering CD8$^+$ T cells are crucial effector cells targeted by ICB and higher MHC-I expression is associated with better ICB response, we next explored whether the therapeutic efficacy of ICB can be sensitized by tumoral GLDC-

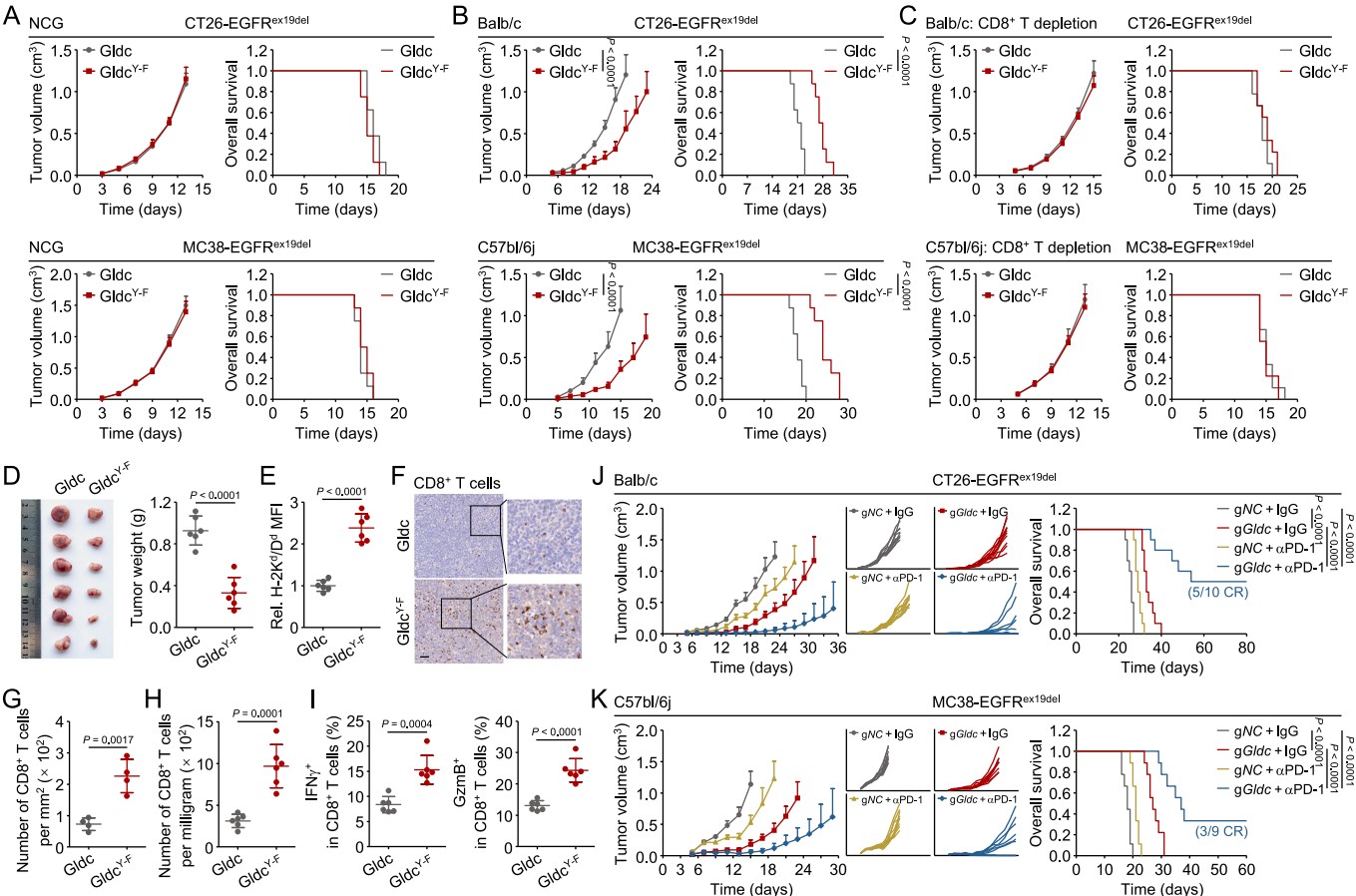

**Figure 7. GLDC-deficiency sensitizes the antitumor effects of PD-1 blockade therapy.**

(A–C) Effects of GLDC mutations on tumor growth in mice. Gldc-deficient (gGldc) CT26-EGFR[ex19del] or MC38-EGFR[ex19del] cells (5 × 10[5]) reconstituted with mouse wild-type Gldc or Gldc[Y998F/Y1013F] (Gldc[Y-F]) were subcutaneously injected into the indicated mice. On day 3 after tumor cell inoculation, tumor sizes were measured every 2 days by caliper. Mice were sacrificed when the tumor size was bigger than 15 mm of the mean tumor diameter, tumor volume exceeded 2000 mm[3], or the tumor had ulcers with a diameter of 10 mm. Graph shows mean ± SEM, n = 8 (A, B), n = 9 (C). Data were analyzed using two-way ANOVA with GraphPad Prism 8. Kaplan–Meier survival curves and corresponding log-rank (Mantel–Cox) tests were used to evaluate the statistical differences between groups in survival studies. There is a significant difference when the P < 0.05. (D) GLDC[Y993F/Y1008F] inhibits tumor growth. The indicated cells were subcutaneously injected into Balb/c mice. Tumor-bearing mice were euthanized on day 18, and then the tumor tissues were separated from the mice. Tumor weights were measured by an Analytical Balance. Graph shows mean ± SEM, n = 6. Data were analyzed using a student's unpaired t-test with GraphPad Prism 8. (E) Effects of GLDC mutations on MHC-I surface expression of tumor cells in vivo. The indicated cells were subcutaneously injected into Balb/c mice. Tumor-bearing mice were euthanized on day 18, and then the tumor tissues were separated from the mice. Tumor cells were isolated from the indicated tumor tissues, stained with the indicated antibodies and analyzed by flow cytometry. Graph shows mean ± SEM, n = 6 independent samples. Data were analyzed using a student's unpaired t-test with GraphPad Prism 8. (F, G) GLDC[Y993F/Y1008F] increases the number of CD8[+] T cells in tumor tissues. The indicated cells were subcutaneously injected into Balb/c mice. After 18 days, tumor-bearing mice were euthanized, and tumor tissues were analyzed. Representative images from IHC staining of CD8 in tumor sections were shown (F). Scale bar, 100 μm. Quantitative analysis of IHC data of CD8-positive cells were shown (G). Graph shows mean ± SEM, n = 4 independent samples. Data were analyzed using a student's unpaired t-test with GraphPad Prism 8. (H, I) GLDC[Y993F/Y1008F] increases the number of CD8[+] T cells in TME. TILs were isolated from the CT26-EGFR[ex19del] tumor tissues in Fig. 7D. TILs were stained with the indicated antibodies and analyzed by flow cytometry. Graph shows mean ± SEM, n = 6 independent samples. Data were analyzed using a student's unpaired t-test with GraphPad Prism 8. (J, K) GLDC-deficiency sensitizes the antitumor effects of PD-1 blocking antibody. Control (gNC) or Gldc-deficient (gGldc) CT26-EGFR[ex19del] or MC38-EGFR[ex19del] cells (5 × 10[5]) were subcutaneously injected into Balb/c or C57bl/6j mice, respectively. On day 7 after tumor cell inoculation, mice were intraperitoneally injected with IgG or anti-PD-1 (100 μg per mouse) every 3 days. Tumor sizes were measured every two days by caliper from day 5. Mice were sacrificed when the tumor size was bigger than 15 mm of the mean tumor diameter, tumor volume exceeded 2000 mm[3], or the tumor had ulcers with a diameter of 10 mm. Graph shows mean ± SEM, n = 10 (J), n = 9 (K). Data were analyzed using two-way ANOVA with GraphPad Prism 8. Kaplan–Meier survival curves and corresponding log-rank (Mantel–Cox) tests were used to evaluate the statistical differences between groups in survival studies. There is a significant difference when the P < 0.05. CR complete response. Source data are available online for this figure.

deficiency. We subcutaneously injected the control or GLDC-deficient CT26-EGFR[ex19del] or MC38-EGFR[ex19del] cells in Balb/c or C57bl/6j mice, respectively, and treated them with PD-1 blocking antibody. As expected, administration of PD-1 blocking antibody in mice inhibited tumor growth, eliminated tumors in five out of

ten injected with GLDC-deficient CT26-EGFR[ex19del] cells and three out of nine injected with GLDC-deficient MC38-EGFR[ex19del] cells (Fig. 7J,K). Taken together, these results suggest that GLDC-deficiency sensitizes the antitumor effects of PD-1 blocking antibody.

# Discussion

Low immunogenicity of TME due to inadequate MHC-I antigen presentation is a common mechanism (Zhang et al, 2022). Understanding the cellular and molecular basis responsible for the inadequate MHC-I antigen presentation in TME is important to design better combination immunotherapy strategies to enhance the efficacy of ICB therapy. In this study, we show that the glycine metabolism key enzyme GLDC functions as an inhibitor of MHC-I expression independent of its enzymatic activity. Upon EGFR activation, GLDC is phosphorylated and subsequently translocated into the nucleus. The nuclear GLDC hijacks the STAT1 co-activator SMARCE1 to inhibit the binding of STAT1 to promoter regions of *IRF1* and *NLRC5* genes, and the GLDC/SMARCE1/STAT1 complex also recruits DMAP1/DNMT1 to induce promoter DNA hyper-methylation of *IRF1* and *NLRC5* genes, resulting in transcriptional inhibition of the downstream MHC-I genes. Genetic ablation of GLDC increases the levels of MHC-I in tumor cells, improves the functions of tumor-specific CD8[+] T cells in TME, and sensitizes the antitumor effects of anti-PD-1 therapy. Our findings uncover the mechanism by which the nuclear GLDC antagonizes MHC-I gene expression in an enzymatic activity-independent manner, and validate GLDC as a potential target for ICB-based combination immunotherapy.

Cancer cells often reprogram metabolic enzymes to support their own malignant transformation. Metabolic enzymes have been reported to be involved in tumor initiation, development, metastasis and immune escape (Guo et al, 2023; Lai et al, 2024; Liu et al, 2020; Wang et al, 2019; Yang et al, 2012). We used a CRISPR screening method to determine whether metabolic enzymes are involved in the regulation of MHC-I antigen presentation. This effort led to the identification of GLDC, which acted as a negative regulator of MHC-I surface expression. GLDC is a mitochondrial pyridoxal 5′-phosphate (PLP)-dependent enzyme that catalyzes the first and rate-limiting step in glycine catabolism (Liu et al, 2021). Genetic mutation of GLDC induces glycine accumulation, leading to nonketotic hyperglycinemia (Bravo-Alonso et al, 2017). Hyperactive GLDC promotes tumor growth by increasing the flux of glycine catabolism in tumor cells (Liu et al, 2021). However, the function of GLDC in tumor immune evasion is unknown. Our experiments indicate that GLDC negatively regulates MHC-I surface expression in EGFR-activated tumor cells and promotes tumor cells to escape CD8[+] T cell-mediated immunosur-veillance by a mechanism independent of its enzymatic activity.

MHC-I surface expression can be affected by many factors, such as pre-translational (e.g., epigenetic modulation, transcriptional regulation and RNA processing) and post-translational (e.g., internalization and degradation) (Chen et al, 2023; Dersh et al, 2021; Liu et al, 2020). Our experiments indicated that GLDC negatively regulated the expression of MHC-I genes (e.g., *HLA-A*, *HLA-B*, *HLA-C*, and *B2M*), but did not affect MHC-I internaliza-tion and degradation. These results suggest that GLDC regulates MHC-I levels at the mRNA but not protein level. The transcription of MHC-I genes is primarily regulated by transcription factors STAT1 and NF-κB (Bell and Zou, 2024; Sykulev, 2023; Wang et al, 2024). It has been shown that activated STAT1 induces expression of transcription factors IRF1 and NLRC5, which ultimately induce transcription of MHC-I genes (Wei et al, 2017). GLDC-deficiency induced GAS (STAT1-targeted) activation but not NF-κB. The mRNA levels of the *IRF1* and *NLRC5* genes were increased in

GLDC-deficient cells. Upregulation of mRNA levels of MHC-I genes in GLDC-deficient cells was reversed by the knockout of STAT1. These results suggest that GLDC regulates transcription of MHC-I genes through the STAT1-IRF1/NLRC5 axis. Phosphory-lated STAT1 forms homodimers, which translocate to the nucleus and bind to the conserved GASs on the promoters to initiate the transcription of these downstream target genes (Villarino et al, 2017). Our experiments showed that GLDC-deficiency induced the binding of STAT1 to promoter regions of their target genes, but did not affect the phosphorylation and nuclear aggregation of STAT1.

Our and others' previous studies have demonstrated that GLDC localizes to mitochondria and cytoplasm, and its catalyzed glycine catabolism is active in mitochondria (Liu et al, 2021). However, here we found that GLDC might be involved in regulating a process that occurs in the nucleus. Metabolic enzymes in the mitochondria or cytoplasm have been reported to be reprogrammed into the nucleus to regulate gene transcription (Huang et al, 2023; Liu et al, 2021; Yang et al, 2012). In our experiments, we found that GLDC was accumulated in the nucleus after EGFR activation. Mechanistically, in EGFR-activated cells, GLDC was phosphorylated by SRC at Y993 and Y1008 sites, and subsequently captured by RanGDP/NTF2 nuclear import system and transported into the nucleus. Knockout of SRC or Ran abolished EGFR activation-mediated GLDC nuclear translocation. L998/V999 of GLDC is important for its interaction with Ran. EGFR activation induced the aggregation of wild-type GLDC but not GLDC[Y993F/Y1008F] or GLDC[L998A/V999A] in the nucleus. Reconstitution of GLDC[Y993F/Y1008F] or GLDC[L998A/V999A] in GLDC-deficient cells failed to reverse the upregulation of surface expression of MHC-I. These results suggest that EGFR activation-induced phosphorylation and nuclear translocation of GLDC mediates inhibition of MHC-I surface expression.

In a screen for potential associated proteins of nuclear GLDC, we identified SMARCE1, which is part of the large ATP-dependent chromatin remodeling complex SWI/SNF and involved in tran-scriptional activation and repression of select genes by chromatin remodeling (Mashtalir et al, 2018). Our experiments showed that SMARCE1 was associated with GLDC in the nucleus. Notably, their interactions were independent of the SWI/SNF complex. Knockout of SMARCE1 abolished upregulation of the binding of STAT1 to *IRF1* promoter region, the mRNA levels of *IRF1* and *NLRC5* genes and the surface expression of MHC-I in GLDC-deficient cells, suggesting that GLDC regulates MHC-I expression through SMARCE1. In our experiments, we also found that SMARCE1 was phosphorylated by SRC at Y73 and Y170 sites after EGFR activation. Phosphorylation of SMARCE1[Y73] and SMARCE1[Y170] enhances its stability and induces its interaction with GLDC, respectively. Upregulation of the binding of STAT1 to the *IRF1* promoter region, the mRNA levels of *IRF1* and *NLRC5* genes, and the surface expression of MHC-I in SMARCE1-deficient cells were abrogated in the cells reconstituted by wild-type SMARCE1, and dramatically enhanced in SMARCE1[Y170F]-reconstituted cells. These results suggest that SMARCE1 promotes STAT1-triggered tran-scription of downstream genes, while is inhibited by the interaction of GLDC with SMARCE1 in the nucleus.

Previous studies have demonstrated that SMARCE1 can regulate gene transcription by binding directly to transcription factors (Sokol et al, 2017). Our experiments further indicate that SMARCE1 is a co-activator of STAT1. SMARCE1 was associated with STAT1, and it was not affected by GLDC-deficiency. GLDC also interacted with STAT1,

but it was abolished by SMARCE1-deficiency. These results suggest that SMARCE1 acts as a bridge to induce the formation of GLDC/SMARCE1/STAT1 complex. Mutation of SMARCE1 I206/L207/V218 to alanine, which blocks STAT1 interaction, has no marked effects on STAT1 binding to the IRF1 promoter region. This suggests that SMARCE1 promotes STAT1 binding to downstream target gene promoters through direct interaction with STAT1. SMARCE1 contains an N-terminal high-mobility group (HMG) DNA-binding domain (St Pierre et al, 2022). GLDC-deficiency or mutation of SMARCE1$^{Y170}$ to phenylalanine induced the binding of SMARCE1 to the IRF1 promoter region, which was abolished by knockout of STAT1 or blocking the interaction of SMARCE1 with STAT1. The SMRCAE1$^{ΔHMG}$ truncation mutant was failed to interact with GLDC, but did not affect the interaction between SMARCE1 and STAT1. The binding of SMARCE1$^{ΔHMG}$ to the *IRF1* promoter region was significantly weaker than SMARCE1$^{Y170F}$, and SMARCE1$^{ΔHMG}$ failed to enhance the binding of STAT1 to the *IRF1* promoter region, suggesting that DNA-binding activity is required for SMARCE1 as a co-activator of STAT1. Moreover, the HMG box contains three α helix structures (A: aa72-87, B: aa93-105, C: aa109-127) that determine the DNA-binding activity of HMG proteins (Stros et al, 2007). Our results showed that GLDC interacted with the B α helix structure of the HMG domain of SMARCE1. Taken together, these results suggest that nuclear GLDC, as a co-repressor of STAT1, hijacks the HMG domain of SMARCE1 to suppress the activity of SMARCE1 as a co-activator of STAT1, leading to STAT1-triggered transcriptional repression and ultimately inhibition of MHC-I antigen presentation.

Our experiments also suggest that DMAP1/DNMT1 mediates another mechanism responsible for GLDC-mediated inhibition of MHC-I expression. DNMT1 is an essential DNA methyltransferase that catalyzes the transfer of methyl groups to CpG islands in DNA and mediates transcriptional repression (Rountree et al, 2000). DMAP1 is a DNMT1-associated protein that can recruit DNMT1 to form a repressive transcription complex (Rountree et al, 2000). Our results demonstrate that nuclear GLDC recruits DNMT1 to the *IRF1* promoter through its interaction with DMAP1, thereby mediating DNA methylation of the *IRF1* promoter, and DMAP1/DNMT1-induced promoter DNA hypermethylation inhibits the transcription of *IRF1* and *NLRC5* genes. In addition, GLDC-deficiency impaired the upregulation of mRNA levels of *IRF1* and *NLRC5* genes induced by inhibition of DNMTs, and the mRNA levels of *IRF1* and *NLRC5* genes in DAC-treated cells were lower than in GLDC-deficient cells, suggesting that DMAP1/DNMT1-mediated transcriptional repression of *IRF1* and *NLRC5* genes is part of the inhibition function induced by GLDC. SMARCE1-deficiency downregulated the levels of *IRF1* promoter DNA methylation and the binding of DNMT1 to the *IRF1* promoter region, which was reversed by reconstitution of wild-type SMARCE1 but not SMARCE1$^{Y170F}$ in those cells, suggesting that recruiting DNMT1 to the *IRF1* promoter region by GLDC/DMAP1 is also dependent on the interaction of GLDC with SMARCE1. Furthermore, DMAP1-deficiency or DNMT1-deficiency did not affect the binding of STAT1 to the IRF1 promoter, suggesting that the DMAP1/DNMT1-induced promoter DNA hypermethylation does not affect the binding of STAT1 to the promoter regions of downstream target genes. Previous studies have demonstrated that DNA methylation suppresses gene transcription through two main mechanisms, directly interfering with the binding of transcription factors to gene promoters and indirect recruitment of transcriptional repressors by 5mC-binding proteins (Jones, 2012). Our observation

aligns with previous studies that DNMT1 and DNMT3b cooperate to mediate *IRF8* promoter DNA methylation, and the DNA methylation inhibits STAT1 activity without blocking its binding to the GAS element of *IRF8* promoter (McGough et al, 2008). Further mechanisms reveal that *IRF8* promoter methylation enables methyl-CpG binding domain protein 1 (MBD1) to recruit repressor PIAS1, thereby inhibiting STAT1 function (McGough et al, 2008). Therefore, we guess that *IRF1* promoter methylation might also lead to binding of methyl-CpG binding proteins that recruit repressors to inhibit STAT1 function. Taken together, our experiments suggest that nuclear GLDC recruits DMAP1/DNMT1 to the promoter region of STAT1-targetted genes after forming a complex with SMARCE1/STAT1, which leads to promoter DNA hypermethylation and ultimately transcriptional inhibition of those genes.

Based on our results of a study on the mechanisms of GLDC-induced EGFR-activated tumor cells to evade CD8$^+$ T cell antitumor immune responses in an enzymatically independent manner, we propose a model for the regulation of MHC-I expression by reprogramming GLDC. Highly active EGFR signaling in cancerous cells activates SRC, which mediates phosphorylation of GLDC at Y993 and Y1008, leading to its capture and transport into the nucleus by the RanGDP/NTF2 nuclear input system. SMARCE1 is a co-activator of STAT1. The nuclear GLDC hijacks SMARCE1 to inhibit the binding of STAT1 to promoter regions of *IRF1* and *NLRC5* genes, and the GLDC/SMARCE1/STAT1 complex also recruits DMAP1/DNMT1 to induce promoter DNA hypermethylation of *IRF1* and *NLRC5* genes, resulting in transcriptional inhibition of downstream MHC-I genes. Genetic ablation of GLDC restores the co-activator activity of SMARCE1 and removes the promoter DNA hypermethylation of *IRF1* and *NLRC5* genes, thereby inducing transcription of *IRF1* and *NLRC5* genes and increasing tumor MHC-I antigen presentation (Fig. EV1). Therefore, nuclear GLDC suppresses tumor MHC-I antigen presentation via two complementary mechanisms. Consistent with this model, GLDC phosphorylation at Y993/Y1008 mediates tumor immune evasion by reducing MHC-I presentation, compromising CD8$^+$ T cell function in TME, and enabling escape from T cell immunosurveillance. In this context, GLDC promotes tumor growth in an enzyme-dependent manner and promotes tumor immune escape in an enzyme-independent manner. Targeting GLDC may increase the efficacy of cancer immunotherapy of PD-1 blockade. Consistently, GLDC-deficiency sensitizes the antitumor effects of PD-1 blockade therapy. In addition, there are numerous types of EGFR mutations that mediate EGFR activation, such as exon 19 deletion mutation, L858R, T790R, and exon 20 insertions, and our experiments primarily used the EGFR$^{ex19del}$ mutant to simulate EGFR activation. Considering that SRC is a commonly downstream effector of EGFR signaling, which phosphorylates GLDC to inhibit transcription of MHC-I upon EGFR activation, our conclusions likely hold for other types of EGFR activation mutations capable of activating SRC.

EGFR is a common therapeutic target for cancer patients. Clinically, EGFR blockade therapy has benefited a large proportion of patients with EGFR-activated cancers (Remon et al, 2018). Blockade of EGFR also improves responsiveness to PD-1 blockade in EGFR-activated cancers (Sugiyama et al, 2020). However, the acquired resistance induced by EGFR tyrosine kinase inhibitor (EGFR-TKI) therapy has become an unsolvable problem (Passaro et al, 2021; Remon et al, 2018). For patients with EGFR-activated tumors to achieve durable remission, developing new EGFR-targeted drugs that circumvent acquired resistance has become a priority. For example,

EGFR proteolysis-targeting chimera (EGFR-PROTAC) has been developed to target EGFR degradation by bridging EGFR with the ubiquitin-proteasome system, and EGFR antibody-drug conjugate (EGFR-ADC) has been developed to achieve cell-specific cytotoxicity through antibody-mediated delivery of apoptosis-inducing agents to EGFR-positive cells (Hong et al, 2022; Kargbo, 2022; Ma et al, 2024; Swain et al, 2023). Systemic GLDC inhibition is thought to cause nonketotic hyperglycinemia, thereby the direct use of GLDC inhibitors as oncology therapeutics is debatable (Coughlin et al, 2017). Considering the dual functions of GLDC in tumor growth and immune evasion in EGFR-activated tumors, the development of an antibody-drug conjugate targeting EGFR-activated cells to deliver GLDC inhibitors will be a promising solution for the therapy of EGFR-activated cancers in the future.

In conclusion, our current study suggests that phosphorylated and nuclear-localized GLDC suppresses the expression of MHC-I genes by hijacking SMARCE1 and recruiting DMAP1/DNMT1 to block STAT1-triggered transcription of *IRF1* and *NLRC5* genes. Genetic ablation of GLDC removes the inhibitory effects on STAT1 and restores tumor MHC-I antigen presentation, thereby improving the function of tumor-specific CD8+ T cells and the efficacy of ICB therapy. Our findings define the underlying mechanisms of how EGFR-activated tumor cells undergo CD8+ T cell immune evasion by reprogramming GLDC. As validated in our study, regulation of these mechanisms would provide new targets for the development of ICB-based combination immunotherapy strategies.

# Methods

### Reagents and tools table

| Reagent/resource | Reference or source | Identifier or catalog number |
|---|---|---|
| **Experimental models** | | |
| C57BL/6J (*M. musculus*) | Hunan SJA Laboratory Animal Co., Ltd | N-0005 |
| BALB/c (*M. musculus*) | Hunan SJA Laboratory Animal Co., Ltd | N-0004 |
| NCG (*M. musculus*) | GemPharmatech Co., Ltd | T001475 |
| **Recombinant DNA** | | |
| pMSCV-hGLDC and mutants | This study | N/A |
| pMSCV-mGldc and mutants | This study | N/A |
| pRK-3'FLAG-hGLDC | This study | N/A |
| pRK-3'HA-hGLDC | This study | N/A |
| pRK-3'HA-hImportins | This study | N/A |
| pRK-3'HA-hRan and mutants | This study | N/A |
| pMSCV-hSMARCE1 and mutant | This study | N/A |
| pMSCV-hDMAP1 and mutants | This study | N/A |
| pRK-3'Myc-hSRC | This study | N/A |
| pRK-3'HA-hSMARCE1 and mutants | This study | N/A |

| Reagent/resource | Reference or source | Identifier or catalog number |
|---|---|---|
| pRK-3'FLAG-hSMARCE1 | This study | N/A |
| pRK-3'HA-hDMAP1 and mutants | This study | N/A |
| pRK-3'FLAG-hDMAP1 | This study | N/A |
| **Antibodies** | | |
| Primary antibodies | This study | Appendix Table S5 |
| Flow antibodies | This study | Appendix Table S6 |
| **Oligonucleotides and other sequence-based reagents** | | |
| gRNA primers | This study | Appendix Table S7 |
| qPCR primers | This study | Appendix Table S8 |
| ChIP-qPCR primers | This study | Appendix Table S9 |
| **Chemicals, enzymes and other reagents** | | |
| Drugs | This study | Appendix Table S4 |
| Recombinant proteins | This study | Appendix Table S4 |
| **Software** | | |
| GraphPad Prism 7 | https://www.graphpad.com/support/prism-7 | |
| CFX ManagerTM version 2.1 | Bio-Rad | N/A |

## Reagents and antibodies

Information on the commercially available reagents used in this study is provided in Appendix Table S4. Information on the commercially available antibodies used in this study is provided in Appendix Table S5. The antibody that specifically recognizes phosphorylated Y993 or Y1008 of GLDC were raised by immunizing rabbits with a synthetic peptide of human GLDC ($_{989}$IDDI(Y-p)GDQH$_{997}$) or human GLDC ($_{1004}$P-Nle-EV(Y-p)ESPF$_{1012}$) by ABclonal Technology (Wuhan), respectively. The antibody that specifically recognizes phosphorylated Y73 or Y170 of SMARCE1 were raised by immunizing rabbits with a synthetic peptide of human SMARCE1 ($_{68}$KPLMP(Y-p)MRYSRK$_{79}$) or human SMARCE1 ($_{165}$EKGEP(Y-p)MSIQP$_{175}$) by ABclonal Technology (Wuhan), respectively.

## Cells

A549 cells and H1299 cells were obtained from the American Type Culture Collection. PC9 cells, CT26 cells, MC38 cells, and LLC cells were provided by Dr. Jinfang Zhang (Wuhan University). B16-OVA cells were provided by Dr. Chen Dong (Tsinghua University). HEK293 cells were originally provided by Dr. Gary Johnson (National Jewish Health, Denver, CO). All cells were cultured in DMEM (GIBCO, Catalog #C11995500) supplemented with 10% FBS (Cell Max, Catalog #SA211.02) and 1% penicillin-streptomycin (GIBCO, Catalog #15140-122) and detected negative for mycoplasma.

## Constructs

Mammalian expression plasmids for Flag-, HA-, or Myc-tagged GLDC, SRC, Ran, Importin α1, Importin α3, Importin α4, Importin

α5, Importin α6, Importin α7, NTF2, SMARCE1, SMARCB1, SMARCD1, SMARCA4, BRD7, BRD9, ACTL6A, STAT1, DMAP1, and their mutants were constructed by standard molecular biology techniques. Guide-RNA plasmids targeting GLDC, STAT1, p65, SRC, RAN, SMARCE1, and DMAP1 were constructed into a lenti-CRISPR V2 vector, which was provided by Dr. Shu-Wen Wu (Wuhan University).

## Transfection

HEK293 cells were transfected by standard calcium phosphate precipitation. A549 and H1299 cells were transfected with Lipofectamine 2000. The empty control plasmid was added to ensure that each transfection receives the same amount of total DNA.

## Flow cytometry analysis

Cells were subjected to stain with the indicated antibodies for 45 min at 4 °C. The cells were analyzed, and data were acquired with BD Fortessa X-20 and FACSDiva 7 software following the exemplified gating strategy for flow cytometry analysis. The data were processed using FlowJo software. The antibodies were used for flow cytometric analyses in this study is provided in Appendix Table S6.

## Surface B2M internalization

Cells were stained with purified anti-B2M antibody for 45 min at 4 °C and then incubated at 4 °C or 37 °C for indicated times before staining for FITC goat anti-mouse IgG for 45 min on ice. The cells were analyzed and data were acquired with BD Fortessa X-20 and FACSDiva 7 software following the exemplified gating strategy for flow cytometry analysis. The data were processed using FlowJo software.

## CRISPR-Cas9 knockout

Double-stranded oligonucleotides corresponding to the target sequences were cloned into the Lenti-CRISPR-V2 vector, which were co-transfected with the packaging plasmids into HEK293 cells. Two days after transfection, the viruses were harvested, ultra-filtrated (0.45 μm filter, Millipore) and used to infect cells in the presence of polybrene (8 μg/mL). The infected cells were selected with puromycin (A549: 2 μg/mL, H1299: 2 μg/mL, PC9: 2 μg/mL, CT26: 4 μg/mL, MC38: 3 μg/mL, LLC: 1 μg/mL, B16-OVA: 1.5 μg/mL) for at least 6 days. The information on gRNA sequences is shown in Appendix Table S7.

## CRISPR screening

A total of 107 metabolic enzymes from Sanger Arrayed Human Whole Genome Lentiviral CRISPR Library (Sigma-Aldrich, HSANGERG-1EA) were included in the sgRNA screen (2 sgRNAs per metabolic enzyme gene). Each of the sgRNA was co-transfected with the packaging plasmids into HEK293 cells, respectively. Two days after transfection, the viruses were harvested, ultra-filtrated (0.45 μm filter, Millipore) and used to infect H1299-Cas9 cells in the presence of polybrene (8 μg/mL). The H1299-Cas9 cells were selected with puromycin (2 μg/mL) for 6 days. The indicated cells were stained with the FITC-HLA-ABC antibodies and analyzed by flow cytometry.

## Proliferation assay

Cells ($5 \times 10^4$) were seeded in 6-cm dish for 24 h. Triplicate dishes were seeded for each experimental group. The cells were trypsinized, resuspended in DMEM containing 10% FBS, and counted with a Cellometer (Bio-Rad) every 2 days over a 6-day period.

## Mouse tumor models

Mice were housed with five mice per cage on a 12-h light/dark cycle in a temperature-controlled room (23–25 °C) and relative humidity of 40–70% with free access to water and food. Mice were allowed at least 7 days for acclimation before experimentation. All animal use and experimental protocols were carried out in compliance with the Institutional Animal Care and Use Committee guidelines and approved by the Animal Care and Ethics Committee of Wuhan University Medical Research Institute.

Age- and sex-matched Balb/c, C57BL/6j, or NCG (all aged 6–8 weeks) were anaesthetized and subcutaneously injected with the indicated mouse tumor cells ($5 \times 10^5$ in 200 μL PBS). The mice were euthanized when the tumor size was bigger than 15 mm of the mean tumor diameter or tumor volume reaches 2000 mm$^3$ or deemed as died.

## Animal survival studies

For survival studies, mice were sacrificed when the tumor size was bigger than 15 mm of the mean tumor diameter, tumor volume exceeded 2000 mm$^3$, or the tumor had ulcers with a diameter reached 10 mm. Statistical analysis was performed using the GraphPad Prism 8 software. Kaplan–Meier survival curves and corresponding log-rank (Mantel–Cox) tests were used to evaluate the statistical differences between groups in survival studies. There is a significant difference when the $P < 0.05$.

## In vivo mouse CD8$^+$ T cells depletion experiments

For the CD8$^+$ T cells depletion in Balb/c and C57BL/6j mice, mice were intraperitoneally injected with anti-CD8α (A2102, dissolved in PBS, Selleck) every three days, starting on day three before tumor cells inoculation. Five injections were given, with the first dose at 200 μg per mouse and next four dose at 100 μg per mouse. The depletion efficacy was validated by flow cytometry.

## In vivo PD-1 blocking therapy in mouse tumor models

For anti-PD-1 therapy in Balb/c and C57BL/6j mice, mice were intraperitoneally injected with IgG (A2123, 100 μg per mouse, dissolved in PBS, Selleck) or anti-PD-1 (BE0273, 100 μg per mouse, dissolved in PBS, BioXCell) every 3 days, starting on day 7 after CT26-EGFR$^{ex19del}$ (Balb/c mice) or MC38-EGFR$^{ex19del}$ (C57BL/6j mice) cells inoculation. Tumor size and mouse survival were measured every 2 days from day 5.

## Isolation of tumor-infiltrated immune cells

Tumor tissues were separated from the mice and cut into pieces. The tumor tissues were suspended with 2 mL of tumor digestion

buffer (1 × HBSS buffer with 5 mg/mL collagenase II and 0.1% DNase I) and rotated at 37 °C for 1 h. The cell suspension was filtered using a 70-µm filter to obtain a single-cell suspension. The lymphocytes were isolated by density-gradient centrifugation using 40 and 70% Percoll (GE). The TILs were stained using fluorescently labeled antibodies for different markers. Cells were analyzed and data were acquired with BD Fortessa X-20 and FACSDiva 7 software following the exemplified gating strategy for flow cytometry analysis. The data were processed using FlowJo software.

## Immunohistochemistry (IHC) staining

IHC staining was performed as previously described (Liu et al, 2021). In brief, the slides were deparaffinized in xylene, and rehydrated sequentially in 100, 95, and 75% ethanol for 5 min. The antigen retrieval was performed by heating slides in a microwave for 30 min in sodium citrate buffer (pH 6.0) or 0.5 mm EDTA buffer (pH 8.0). The sections were cooled down naturally to room temperature and quenched in 3% hydrogen peroxide to block endogenous peroxidase activity. The sections were incubated with antibodies overnight at 4 °C. Next, a secondary biotinylated immunoglobulin G antibody solution and an avidin-biotin peroxidase reagent were added to the slides. After washing with phosphate buffer saline, 3,3′-diaminobenzidine tetrachloride was added to the sections, followed by counterstaining with hematoxylin (Beyotime Biotech). The information and dilutions of antibodies are listed in Appendix Table S3. Signals were imaged with an Aperio VERSA 8 (Leica) multifunctional scanner and quantified with the software Image-Pro Plus 6.0.

## Co-immunoprecipitation and immunoblotting analysis

Cells were lysed in 1 mL of NP-40 lysis buffer (20 mM Tris-HCl, 150 mM NaCl, 1 mM EDTA, 1% Nonidet P-40, 1% Triton X-100, 10 µg/mL aprotinin, 10 µg/mL leupeptin, and 1 mM phenylmethylsulfonyl fluoride, PMSF). For each immunoprecipitation reaction, a 0.4 mL aliquot of lysate was incubated with 0.5–2 µg of the indicated antibody or control IgG and 35 µL of a 1:1 slurry of Protein-G Sepharose at 4 °C for 3 h. The Sepharose beads were washed three times with 1 mL of lysis buffer containing 500 mM NaCl. The precipitates were fractionated by SDS-PAGE, and immunoblotting analysis was performed with the indicated antibodies.

## Cell fractionation assays

Cells were untreated or treated with EGF for the indicated times. The cell fractionation assays were performed by NE-PER Nuclear and Cytoplasmic Extraction Reagents (Thermo, Catalog #78835).

## Quantitative real-time PCR (qPCR)

Total RNA was isolated for qPCR analysis to measure mRNA abundance of the indicated genes. Data shown are the relative abundance of the indicated mRNAs normalized to that of GAPDH or β-actin. The qPCR data were collected with Bio-Rad CFX96 (Version 3.1) and analyzed with Bio-Rad CFX Manager (Version 3.1). Gene-specific primer sequences is listed in Appendix Table S8.

## OT-I and tumor cells co-culture

C57BL/6-Tg (TcraTcrb) 1100Mjb/J mice were provided by Dr. Jinfang Zhang (Wuhan University). OT-I CD8$^+$ T cells were isolated from the spleen of C57BL/6-Tg (TcraTcrb) 1100Mjb/J mice by negative selection magnetic beads (STEM CELL Technologies, Catalog #19853 A). B16-OVA-EGFR$^{ex19del}$ cells were transduced with a lentivector expressing luciferase. OT-I cells were cultured in RPMI 1640 supplemented with 10% FBS, 1% penicillin-streptomycin, 1% non-essential amino acids, 2 mM L-glutamine, 1 mM sodium pyruvate, 55 µM β-mercaptoethanol, IL-2 (400 IU/mL), and 1 µg/ml OVA$_{257-264}$ peptide. On day 7, the activated OT-I cells were cocultured with Luc$^+$ B16-OVA-EGFR$^{ex19del}$ cells for 36 h at the indicated effector-to-tumor ratio (E:T). Supernatant was removed after co-culture, and the residual cells were lysed before luciferase assays. Total luciferase activity in each well was assessed, and relative tumor cell viability was calculated after normalization with control wells containing only tumor cells.

## Recombinant protein purification

To prepare Flag-GLDC and Flag-GLDC$^{Y993F/Y1008F}$, mammalian expression plasmids for Flag-GLDC and Flag-GLDC$^{Y993F/Y1008F}$ were transfected into HEK293 cells. The cells were lysed 24 h after transfection. Flag antibody-conjugated beads were then used for immunoprecipitation for 4 h at 4 °C. The beads were washed three times with lysis buffer. The Flag-tagged GLDC and GLDC$^{Y993F/Y1008F}$ were eluted with 3 × Flag peptides in 250 mM Tris-HCl, pH 8.0. The Flag-tagged GLDC and GLDC$^{Y993F/Y1008F}$ were used for the kinase assay in vitro.

## In vitro kinase assay

The HEK293-purified Flag-tagged GLDC or GLDC$^{Y993F/Y1008F}$ were incubated with commercial GST-SRC (active) in kinase buffer (60 mM HEPES pH 7.5, 5 mM MgCl$_2$, 5 mM MnCl$_2$, 3 µM Na$_3$VO$_4$, 1.25 mM DTT, and 20 µM ATP). The reactions were performed in a total volume of 50 µL at 37 °C for 30 min. The reaction was stopped by adding SDS-PAGE loading buffer and heating to 95 °C for 5 min. The proteins were subjected to immunoblotting analysis with the indicated antibodies.

## GST pulldown assay

GST-tagged Ran proteins were incubated with nucleotide loading buffer (10 mM EDTA, 10 mM MgCl$_2$, 10 mM GTP, or GDP) for 60 min at RT to obtain GTP or GDP-bound form Ran. GST fusion proteins (5 µg) were incubated in 1 mL NP-40 lysis buffer containing prepared cell lysate together with glutathione agarose beads for 2 h at RT. The glutathione agarose beads were then washed four times with 1 mL of lysis buffer containing 500 mM NaCl and subjected to immunoblotting analysis with the indicated antibodies (Lu et al, 2014).

## Chromatin immunoprecipitation (ChIP)

ChIP was performed according to the manufacturer's instructions. Ten million cells were fixed with 1% formaldehyde for 10 min,

quenched with 0.125 M glycine for 5 min at 37 °C and lysed in SDS Lysis Buffer. Cell lysate was sonicated by Bioruptor Pico Sonifier to shear chromatin DNA to a size range of 200–1000 bp. The supernatant was diluted tenfold in ChIP Dilution Buffer and precleared with 60 mL agarose beads for 30 min. The supernatant fraction was immunoprecipitated with the indicated antibodies (2 µg) overnight at 4 °C. The antibody-chromatin complexes were pulled down with protein A agarose/salmon sperm DNA beads (Sigma, Catalog #16-157) for 1 h at 4 °C. The de-crosslinked DNA was subjected to qPCR analysis using specific primers listed in Appendix Table S8 (Liu et al, 2023).

### Methylated DNA immunoprecipitation (MeDIP)

MeDIP was performed as previously described (Thu et al, 2009). In brief, DNA extraction was performed using the QIAamp DNA Mini Kit (Qiagen, Cat. #51304) according to the manufacturer's instructions. About 2 µg of genomic DNA was sonicated by Bioruptor Pico Sonifier (the range of DNA fragments is 200 to 500 bp) and used for MeDIP. MeDIP was performed using a monoclonal antibody against 5mC (Abcam, clone: 33D3, Cat. #ab10805) labeled with magnetic Dynabeads anti-mouse IgG (Invitrogen, Cat. #M-280). Sheared DNA and antibody coupled to the magnetic beads were incubated overnight at 4 °C. DNA combined with beads was washed five times with immunoprecipitation wash buffer, and DNA was eluted from the beads in digestion buffer [10 mmol/L EDTA, 10 mmol/L Tris-HCl (pH 8.0), 0.5% SDS, 50 mmol/L NaCl] with proteinase K at 50 °C for 30 min. Five percent input DNA was used as a control. The sequences of the primers used for ChIP-qPCR are listed in Appendix Table S9.

### Mass spectrometry

HEK293/EGFR cells ($1 \times 10^8$) were transfected with Flag-tagged human GLDC for 24 h and then treated with or without EGF (100 ng/ml) for 6 h. Flag-tagged GLDC was immunoprecipitated with anti-Flag and desalted, then analyzed by mass spectrometry. Mass spectrometry analysis was performed as previously described by SpecAlly (Wuhan) Life Science and Technology Company (Liu et al, 2021; Liu et al, 2023).

### Confocal microscopy

A549 cells were untreated or treated with EGF for 6 h or transfected with the indicated plasmids for 20 h. The cells were fixed with 4% paraformaldehyde for 15 min, and permeabilized with 0.3% Triton X-100 in PBS for 15 min. The cells were blocked with 5% BSA in PBS and stained with the indicated primary and secondary antibodies. The nuclei were stained with DAPI for 2 min and then washed with PBS three times. The stained cells were observed with a Zeiss LSM880 confocal microscope under a 40 × objective.

### Statistics and reproducibility

Data were analyzed using a student's unpaired *t*-test, multiple *t*-test, or two-way ANOVA with GraphPad Prism 8. The correlation study was analyzed using a Spearman rank correlation test. The number of asterisks represents the degree of significance with respect to *P* values, with the latter presented within each figure or figure legend. All the biochemical experiments, particularly

immunoblotting analysis, were repeated at least two times with similar results. The reproducibility of other experiments is described in the respective figure legends.

## Data availability

All the data supporting the findings of this study are available within the article and its supplementary information files, or can be obtained from the corresponding author upon reasonable request. Source data are included in this submission. No data amenable to large-data repository deposition have been generated in this study.

The source data of this paper are collected in the following database record: biostudies:S-SCDT-10_1038-S44318-025-00557-3.

## Peer review information

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

## Acknowledgements

We thank Dr. Xuan Zhong, Dr. Jie Shi, Dr. Haojian Zhang, Dr. Jinfang Zhang, and members of our laboratory for their technical help and suggestions. This work was supported by grants from the State Key R&D Program of China (2022YFA1304900), the National Natural Science Foundation of China (32188101 and 32300758), the Fundamental Research Funds for the Central Universities (2042022dx0003, 2042025kf0043), the Natural Science Foundation of Wuhan (2024040701010031), and the China Postdoctoral Science Foundation (2022TQ0236 and 2023M732694).

## Author contributions

**Rui Liu**: Conceptualization; Funding acquisition; Investigation; Writing—original draft; Writing—review and editing. **Hui-Fang Li**: Investigation. **Qi Jiang**: Investigation. **Jun-Ge Shi**: Investigation. **Zi-Lun Ruan**: Formal analysis. **Peng Ren**: Investigation. **Yi-Nuo Li**: Investigation. **Hong-Bing Shu**: Conceptualization; Funding acquisition; Writing—original draft; Project administration; Writing—review and editing. **Shu Li**: Conceptualization; Formal analysis; Supervision; Funding acquisition; Writing—original draft; Project administration; Writing—review and editing.

Source data underlying figure panels in this paper may have individual authorship assigned. Where available, figure panel/source data authorship is listed in the following database record: biostudies:S-SCDT-10_1038-S44318-025-00557-3.

## Disclosure and competing interests statement

The authors declare no competing interests.

# Expanded View Figure

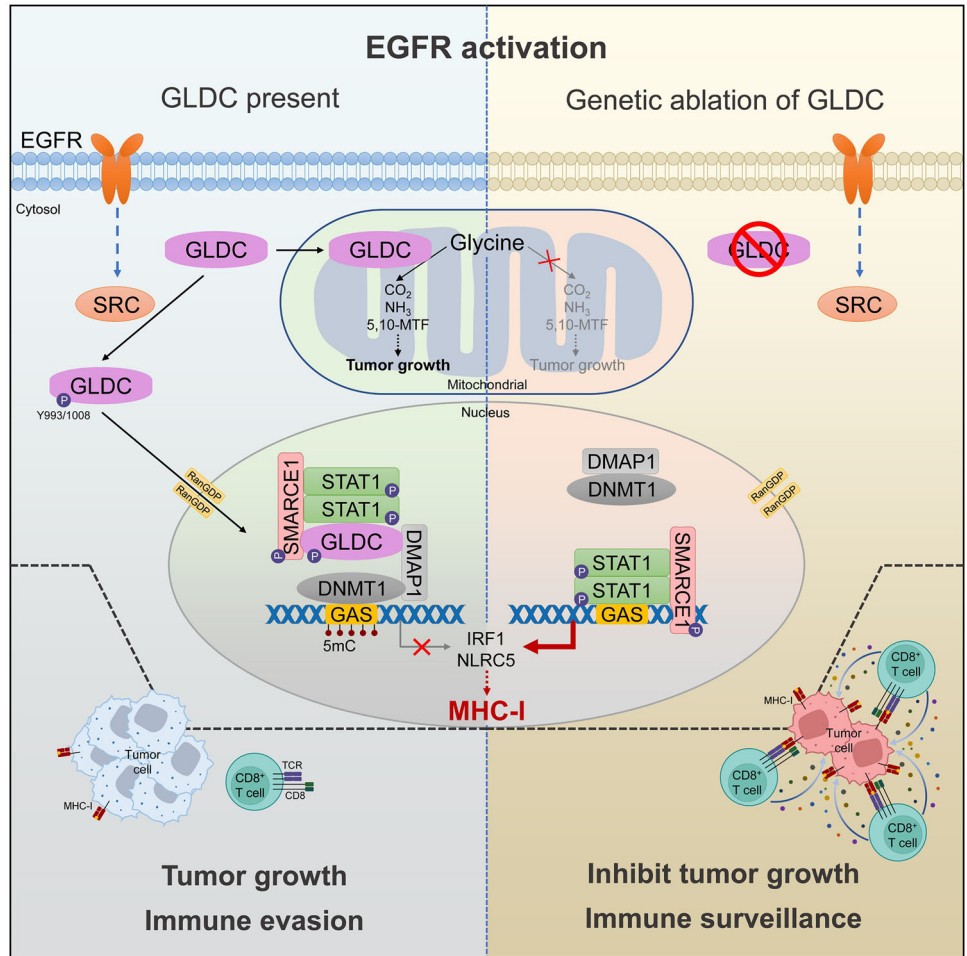

**Figure EV1.** A model on the regulatory of MHC-I antigen presentation by reprogramming GLDC.

