## [Peer Review File · The EMBO Journal]

Nuclear glycine decarboxylase suppresses STAT1-dependent MHC-I and promotes cancer immune evasion

Rui Liu, Hui-Fang Li, Qi Jiang, Jun-Ge Shi, Zi-Lun Ruan, Peng Ren, Yi-Nuo Li, Hong-Bing Shu & Shu Li *Corresponding*

author: Shu Li (shuli@whu.edu.cn)

Review Timeline:

Submission Date:	18th Mar 25
Editorial Decision:	6th May 25
Revision Received:	25th Jun 25
Editorial Decision:	31st Jul 25
Revision Received:	2nd Aug 25
Accepted:	14th Aug 25

Editor: Daniel Klimmeck

Transaction Report:

Dear Dr Li,

Thank you again for the submission of your manuscript (EMBOJ-2025-120829) to The EMBO Journal. Your manuscript was sent to three reviewers with expertise in cancer gene expression control, immunometabolism and tumor biology, and we have now received reports from the all of them, which I enclose below. Based on all information at hand, we have now decided to invite you to revise your work for the EMBO Journal.

As you will see from the experts' reports, the referees acknowledge the analysis and potential interest and value of your findings. However, they also express important issues, which need to be addressed thoroughly to make them supportive of publication in the EMBO Journal. Further, the reviewers raise a number of issues related to the presentation of the findings, statistics applied and overall discussion of related literature, that would need to be conclusively addressed to achieve the level of robustness and clarity needed for The EMBO Journal.

Given the overall interest stated and broader angle of your findings, we are able to invite you to revise your manuscript experimentally to address the referees' comments. I need to stress though that we do require strong support from the referees on a revised version of the study in order to move on to publication of the work.

I would appreciate if you could contact me during the next weeks for exchange e.g. a video call to discuss your perspective on the comments and potential plan for revisions.

Please feel free to contact me if you have any questions or need further input on the referee comments.

When submitting your revised manuscript, please carefully review the instructions below.

Please feel free to approach me any time should you have additional questions related to this.

Thank you for the opportunity to consider your work for publication.

I look forward to your revision.

Kind regards,

Daniel Klimmeck

Daniel Klimmeck, PhD
Senior Editor
The EMBO Journal

Instruction for the preparation of your revised manuscript:

- 1) a .docx formatted version of the manuscript text (including legends for main figures, EV figures and tables). Please make sure that the changes are highlighted to be clearly visible.
- 2) individual production quality figure files as .eps, .tif, .jpg (one file per figure).
- 3) a .docx formatted letter INCLUDING the reviewers' reports and your detailed point-by-point response to their comments. As part of the EMBO Press transparent editorial process, the point-by-point response is part of the Review Process File (RPF), which will be published alongside your paper.
- 4) a complete author checklist, which you can download from our author guidelines ([https://wol-prod-cdn.literatumonline.com/pb-assets/embo-site/Author Checklist%20-%20EMBO%20J-1561436015657.xlsx](https://wol-prod-cdn.literatumonline.com/pb-assets/embo-site/Author%20Checklist%20-%20EMBO%20J-1561436015657.xlsx)). Please insert information in the checklist that is also reflected in the manuscript. The completed author checklist will also be part of the RPF.

6) It is mandatory to include a 'Data Availability' section after the Materials and Methods. Before submitting your revision, primary datasets produced in this study need to be deposited in an appropriate public database, and the accession numbers and database listed under 'Data Availability'. Please remember to provide a reviewer password if the datasets are not yet public (see <https://www.embopress.org/page/journal/14602075/authorguide#datadeposition>).

7) Our journal encourages inclusion of *data citations in the reference list* to directly cite datasets that were re-used and obtained from public databases. Data citations in the article text are distinct from normal bibliographical citations and should directly link to the database records from which the data can be accessed. In the main text, data citations are formatted as follows: "Data ref: Smith et al, 2001" or "Data ref: NCBI Sequence Read Archive PRJNA342805, 2017". In the Reference list, data citations must be labeled with "[DATASET]". A data reference must provide the database name, accession number/identifiers and a resolvable link to the landing page from which the data can be accessed at the end of the reference. Further instructions are available at .

8) At EMBO Press we ask authors to provide source data for the main and EV figures. Our source data coordinator will contact you to discuss which figure panels we would need source data for and will also provide you with helpful tips on how to upload and organize the files.

Numerical data can be provided as individual .xls or .csv files (including a tab describing the data). For 'blots' or microscopy, uncropped images should be submitted (using a zip archive or a single pdf per main figure if multiple images need to be supplied for one panel). Additional information on source data and instruction on how to label the files are available at .

9) We replaced Supplementary Information with Expanded View (EV) Figures and Tables that are collapsible/expandable online (see examples in <https://www.embopress.org/doi/10.15252/embj.201695874>). A maximum of 5 EV Figures can be typeset. EV Figures should be cited as 'Figure EV1, Figure EV2" etc. in the text and their respective legends should be included in the main text after the legends of regular figures.

11) For data quantification: please specify the name of the statistical test used to generate error bars and P values, the number (n) of independent experiments (specify technical or biological replicates) underlying each data point and the test used to calculate p-values in each figure legend. The figure legends should contain a basic description of n, P and the test applied. Graphs must include a description of the bars and the error bars (s.d., s.e.m.).

We realize that it is difficult to revise to a specific deadline. In the interest of protecting the conceptual advance provided by the work, we recommend a revision within 3 months (4th Aug 2025). Please discuss the revision progress ahead of this time with the editor if you require more time to complete the revisions.

Referee #1:

This study reveals that GLDC suppresses MHC-I expression in EGFR-mutant tumors through a non-enzymatic mechanism, providing novel insights into tumor immune evasion. The work elucidates GLDC's nuclear translocation and its interaction with the STAT1 and DMAP1/DNMT1 complex, and validates in animal models that targeting GLDC enhances CD8+ T cell-mediated immune responses and synergizes with anti-PD-1 therapy, demonstrating significant translational value. These findings lay a critical foundation for developing combined immunotherapy strategies against EGFR-mutant tumors. The experimental design is rigorous, and the figures are generally clear, with high clinical relevance. However, several limitations should be addressed:

1. Most data figures are clearly presented, but some images (e.g., confocal images in Figures 4A and 4L; IHC images in Figures 2I and 7F) require higher-magnification views to improve clarity.
2. There are numerous types of EGFR mutations. The experiments primarily utilized models with exon 19 in-frame deletions. The potential impact of other EGFR mutation subtypes (e.g., L858R, T790M, exon 20 insertions) on the conclusions should be explicitly discussed.
3. The mechanistic findings are robustly validated in vitro and in animal models. To strengthen clinical relevance, additional analyses of clinical tissue samples about correlations between GLDC expression, MHC-I levels, CD8+ T cell infiltration, and patient prognosis are recommended.
4. The overall logic is clear, but redundant descriptions in the Discussion and figure legends could be streamlined to emphasize core discoveries.
5. Reference formatting inconsistencies exist (e.g., irregular journal name abbreviations). Please unify citations according to the target journal's guidelines.

Referee #2:

This manuscript by Liu et al. identifies glycine decarboxylase (GLDC) as a novel, non-enzymatic suppressor of MHC-I antigen presentation in EGFR-mutant tumors. Mechanistically, EGFR activation leads to SRC-mediated phosphorylation of GLDC, promoting its nuclear translocation where it sequesters SMARCE1, a STAT1 co-activator, thereby inhibiting STAT1-driven transcription of IRF1 and NLRC5. GLDC also recruits DNMT1 to hypermethylate their promoters, further silencing MHC-I expression. GLDC inhibition restores MHC-I levels, enhances CD8+ T cell responses, and improves anti-PD-1 therapy efficacy. While the findings are novel and the mechanistic insights are compelling, further evidence is necessary to substantiate the authors' claims. If the points below are addressed, this manuscript would be suitable for publication in the EMBO journal.

Major comments

1. The authors claim that the regulatory mechanism involving GLDC is specific to EGFR-mutant cancers, yet much of the experimental work appears to be conducted using the A549 cell line. However, A549 and H1299 have been widely reported to harbor wild-type (WT) EGFR rather than EGFR mutations and constitutively active EGFR signaling (Okabe et al., *Cancer Res.*, 2007; Amann et al., *Cancer Res.*, 2005; Simonetti et al., *J. Transl Med.*, 2010). The authors are encouraged to clarify the EGFR status of these NSCLC cells used in this study and explain how these findings support a mechanism specifically relevant to EGFR-mutant tumors (line 23 and 125-126). If alternative models were used to confirm findings in EGFR-mutant contexts, please include these data or provide further justification.
2. The authors report that HLA-ABC and B2M expression is regulated by EGFR-mediated signaling. However, in Figure 1C, treatment with EGFR inhibitors in A549 (EGFR-WT) and PC9 (EGFR-mutant) cells appears to show no significant difference in relative MFI. To better illustrate any differential effects of Afatinib on NSCLC cells with distinct EGFR statuses, the authors should present the absolute MFI values instead of, or in addition to, relative MFI. This would allow for clearer interpretation of the impact of EGFR signaling on antigen presentation.
3. In Figure 1D, the authors show that cells harboring the EGFR exon 19 deletion (EGFR^{ex19del}) exhibit a significant reduction in H-2Kb /Db expression in the control knockdown group, while GLDC knockdown cells also show a reduction. This result appears counterintuitive, as GLDC is proposed to suppress MHC-I expression. The authors should clarify this observation and further mechanistic insights would also be appreciated.
4. In Figures 1F and 1G, the authors demonstrate that GLDC knockdown leads to increased H-2Kb /Db expression and reduced tumor cell viability in vitro. However, to strengthen the translational relevance of these findings and ensure consistency with the in vitro co-culture results, the authors should include data on tumor growth from an in vivo xenograft model. This would provide

crucial validation that the observed effects of GLDC knockdown on antigen presentation and tumor suppression extend beyond the in vitro setting.

5. In Figures 2I and 7F, the authors present a single representative image of IHC staining. To improve the rigor and interpretability of these results, the authors should provide quantitative analysis of the IHC data, such as positive cell count, along with data from multiple biological replicates.

Minor comments

1. The authors should correct typographical errors found in lines 219, 393, 653, and 709—for example, "lessor decree" should be revised to "lesser degree."
2. The authors have not clearly described the methodology used for the CRISPR screening. Details such as the library used, screening conditions, selection criteria, and analysis pipeline are essential for evaluating the validity and reproducibility of the findings. We recommend the authors provide a more comprehensive explanation of how the screening was performed, either in the main text or supplementary materials.
3. In Figure 1A, the x-axis of the histogram lacks a scale, making it difficult to interpret the data accurately. The authors are advised to include appropriate axis labels and units.
4. In Figure 1A, please include the p-values for the CRISPR screening results. This could be visualized in the form of a volcano plot, for example.
5. The authors use varying labels for the fluorescence intensity, "MFI," and "relative MFI," throughout the manuscript. For clarity and consistency, we recommend standardizing the terminology across all text and figure panels. This will help readers to better follow the data presentation and interpretation.
6. Regarding Figure 5, please clarify in the results description that SMARCE1 is referred to as "CE1" in the figure.

Referee #3:

Tumor immune evasion is a common mechanism that limits the clinical efficacy of ICB. A sufficient presence of tumor-infiltrating immune cells is essential for patient response to ICB. One strategy to enhance tumor "hotness" involves boosting tumor antigen presentation. However, the regulatory mechanisms that control antigen presentation and could serve as therapeutic targets remain poorly understood. In this study, Liu et al. present extensive data supporting a model in which the metabolic enzyme GLDC suppresses antigen presentation through a novel SMARCE1/STAT1- and DMAP1-dependent signaling pathway. Through a CRISPR screen, the authors discovered that GLDC deletion led to increased MHC-I surface expression. Follow-up biochemical assays revealed that this effect is independent of GLDC's mitochondrial enzymatic activity. Instead, GLDC acts via a SRC-dependent tyrosine phosphorylation mechanism within the nucleus.

GLDC was shown to interact with SMARCE1 and DMAP1, acting to suppress STAT1 promoter binding and increase promoter methylation, thereby downregulating MHC-I gene transcription. Functional rescue experiments using specific mutants confirmed the importance of these molecular interactions in regulating MHC-I expression. Furthermore, in vivo studies demonstrated that GLDC deletion enhances ICB efficacy in a manner dependent on tumor antigen presentation and TIL infiltration. Overall, this is a compelling study that offers novel mechanistic insights into the regulation of antigen presentation, with potential implications for cancer immunotherapy. The data are extensive and generally well presented. However, several concerns remain regarding cell line choices and the mechanistic interpretations, as outlined below.

1. The rationale for using the H1299 cell line in the initial CRISPR screen should be provided, particularly given the focus on MHC-I expression.
2. Additional hits besides GLDC are evident in Figure 1A. The authors should acknowledge this and justify their decision to focus on GLDC.
3. In Figure 1C, Afatinib appears to further increase B2M levels in gGLDC compared to gNC cells. Please recheck the statistical analysis.
4. Tumor size and weight data should be included for completeness in Figure 1F.
5. In Figure 1H, clarify whether human and mouse GLDC proteins are conserved sufficiently to justify the use of human GLDC to rescue mouse GLDC knockout.
6. In Figure 2G, tumors from WT-GLDC rescue groups should also be included for proper comparison.
7. Specify in Figures 2I-J whether CD8⁺ cells were sorted from CD45⁺ populations.
8. In Figures 3K-L, it should be clarified whether GLDC directly binds to STAT1 or chromatin.
9. Total GLDC blots are missing in Figures 4D, 4E, and 4G and should be included.
10. An HA blot in the HA-IP sample is missing in Figure 4I and needs to be added.
11. Figure 4 suggests that tyrosine-phosphorylated GLDC blocks STAT1 binding to MHC-I gene promoters. Clarify whether this is due to increased affinity of GLDC for the promoters, or mediated through other transcription factors.
12. The rationale for using HEK293 cells in Table EV3, rather than the tumor lines used in earlier figures, should be explained. Additionally, since the function appears nucleus-specific, a nuclear GLDC interactome would be more appropriate.
13. In Figure EV6C, provide evidence that there are at least two pools of SMARCE1—one associated with SWI/SNF complexes and another with phosphorylated GLDC. Also, indicate whether GLDC remains phosphorylated upon nuclear translocation.
14. From the ChIP assays, clarify whether pSMARCE1^{Y170} co-localizes with GLDC at MHC-I promoters, or whether their interaction occurs primarily in chromatin-free nuclear fractions.
15. Further clarification is needed on how DMAP1 and SMARCE1 binding to GLDC influence each other, as shown in Figure 6.
16. In Figure 6K, the authors should speculate on why CpG methylation does not appear to impair STAT1 binding to the

promoters in this context.

17. In Figure 6N, it remains unclear how SMARCE1 functions in DMAP1-altered cells. It should also be explored whether DMAP1 induces global CpG methylation changes that secondarily impact MHC-I promoter activity.

Response Letter

We would like to thank the reviewers for the insightful suggestions and comments, which have helped to improve the manuscript. We have performed additional experiments and textual revisions to address all the concerns. Followings are our point-to-point responses. The changes are highlighted in the revised manuscript.

Referee #1:

1. Most data figures are clearly presented, but some images (e.g., confocal images in Figures 4A and 4L; IHC images in Figures 2I and 7F) require higher-magnification views to improve clarity.

Reply: We have now replaced the original low-resolution figures with higher-definition images (Figures 4A, 4L, 2I and 7F) and added higher-magnification views (Figure 2I and 7F).

2. There are numerous types of EGFR mutations. The experiments primarily utilized models with exon 19 in-frame deletions. The potential impact of other EGFR mutation subtypes (e.g., L858R, T790M, exon 20 insertions) on the conclusions should be explicitly discussed.

Reply: Our experiments primarily used the EGFR^{ex19del} mutant to simulate EGFR activation. Considering that SRC is a commonly downstream effector of EGFR signaling, which phosphorylates GLDC to inhibit transcription of MHC-I upon EGFR activation, our conclusions likely hold for other types of EGFR activation mutation capable of activating SRC. We have now added the discussion in the revised manuscript (page 30).

3. The mechanistic findings are robustly validated in vitro and in animal models. To strengthen clinical relevance, additional analyses of clinical tissue samples about correlations between GLDC expression, MHC-I levels, CD8⁺ T cell infiltration, and patient prognosis are recommended.

Reply: Previous studies have demonstrated that GLDC is commonly overexpressed in various human tumor cells, and its expression levels are negatively correlated with patient prognosis (Zhang et al., 2012, Cell; Alptekin et al., 2019, Oncogene). MHC-I levels and CD8⁺ T cell infiltration are generally associated with better clinical outcomes (Bell et al., 2024, Annu Rev Immunol; Zhang et al., 2022, Trends Immunol; Roerden et al., 2025, Nat Rev Immunol; Schaafsma et al., 2021, Br J Cancer). To evaluate the clinical relevance of GLDC expression in relation to MHC-I levels and

CD8⁺ T cell infiltration, we analyzed TCGA datasets. In lung squamous cell carcinoma (LUSC), although GLDC was highly expressed in tumors and high GLDC expression correlated with poor prognosis, no significant correlation was observed with MHC-I levels or CD8⁺ T cell infiltration.

Based on our results, phosphorylation of GLDC^{Y993/Y1008} is critical for its role in regulation of MHC-I expression and CD8⁺ T cell immunity. To investigate the mechanism, we developed two rabbit polyclonal antibody targeting Y993-phosphorylated GLDC (pGLDC^{Y993}) and Y1008-phosphorylated GLDC (pGLDC^{Y1008}). These antibodies successfully detected phosphorylation of GLDC in the immunoprecipitated samples (see Fig 4E-G). However, in the whole-cell lysate samples, we observed weak signal specificity, with nonspecific bands interfering with detection. Consequently, these antibodies are unsuitable for IHC analysis of GLDC phosphorylation in clinical tissue samples. This limitation hinders further exploration of clinical relevance of GLDC phosphorylation, including its potential correlations with MHC-I levels, CD8⁺ T cell infiltration, and patient prognosis in tumor samples.

4. The overall logic is clear, but redundant descriptions in the Discussion and figure legends could be streamlined to emphasize core discoveries.

Reply: Following the reviewer's suggestion, we have now streamlined the discussion and figure legends (see the discussion and figure legends parts).

5. Reference formatting inconsistencies exist (e.g., irregular journal name abbreviations). Please unify citations according to the target journal's guidelines.

Reply: We have now corrected it.

Referee #2:

Major comments

1. *The authors claim that the regulatory mechanism involving GLDC is specific to EGFR-mutant cancers, yet much of the experimental work appears to be conducted using the A549 cell line. However, A549 and H1299 have been widely reported to harbor wild-type (WT) EGFR rather than EGFR mutations and constitutively active EGFR signaling (Okabe et al., Cancer Res., 2007; Amann et al., Cancer Res., 2005; Simonetti et al., J. Transl Med., 2010). The authors are encouraged to clarify the EGFR status of these NSCLC cells used in this study and explain how these findings support a mechanism specifically relevant to EGFR-mutant tumors (line 23 and 125-126). If alternative models were used to confirm findings in EGFR-mutant contexts, please include these data or provide further justification.*

Reply: A549 and H1299 cells harbor wild-type EGFR and constitutively active EGFR signaling, while PC9 cells harbor the EGFR^{ex19del} mutation. We have now clarified the EGFR status of these NSCLC cells used in this study (page 7).

We primarily used A549 cells (wild-type EGFR) to verify the GLDC regulatory process downstream of EGFR activation. To further validate our findings, we transduced the mouse tumor cells (CT26, MC38, B16-OVA and LLC) with human EGFR^{ex19del} (EGFR exon 19 deletion mutation) to mimic the constitutively EGFR activation *in vitro* (Fig 1D-H; Fig 3E, H, L; Appendix Figure S4F-G; Appendix Figure S6J). The same EGFR-activated models were used to establish mouse tumor models (Fig 2, Fig 7), demonstrating that GLDC-mediated MHC-I suppression is reproducible across systems. Our initial description of “EGFR-mutant” may have been misleading, as the core mechanism was validated in both wild-type EGFR (A549) and engineered EGFR-activated models. We have now replaced this term with “EGFR-activated” throughout the revised manuscript.

Various previous studies have similarly used wild-type EGFR-constitutive activating cells to study EGFR-driven tumorigenesis and then validated the findings in models of engineered EGFR activation (Yang et al., 2011, Nature; Yang et al., 2012, Cell; Xia et al., 2017, Journal of Experimental Medicine; Lee et al., 2018, Molecular Cell; Sugiyama et al., 2020, Science immunology; Liu et al., 2023, Nature Cell Biology; Wu et al., 2023, Nature Cell Biology; Zhu et al., 2025, Cell Metabolism).

2. *The authors report that HLA-ABC and B2M expression is regulated by EGFR-mediated signaling. However, in Figure 1C, treatment with EGFR inhibitors in A549 (EGFR-WT) and PC9 (EGFR-mutant) cells appears to show no significant difference in relative MFI. To better illustrate any differential effects of Afatinib on NSCLC cells with distinct EGFR statuses, the authors should present the absolute MFI values instead of, or in addition to, relative MFI. This would allow for clearer*

interpretation of the impact of EGFR signaling on antigen presentation.

Reply: Following the reviewer's suggestion, we have now added the absolute MFI values to clearly present the data (see new Appendix Fig S1E).

3. In Figure 1D, the authors show that cells harboring the EGFR exon 19 deletion (EGFR^{ex19del}) exhibit a significant reduction in H-2Kb /Db expression in the control knockdown group, while GLDC knockdown cells also show a reduction. This result appears counterintuitive, as GLDC is proposed to suppress MHC-I expression. The authors should clarify this observation and further mechanistic insights would also be appreciated.

Reply: As shown in Figure 1D, EGFR^{ex19del} inhibited MHC-I surface expression, and GLDC-deficiency impaired but not abolished the inhibitory effect caused by overexpression of EGFR^{ex19del}. These results suggest GLDC plays a contributory role in EGFR-mediated MHC-I suppression, and EGFR can also down-regulate MHC-I through GLDC-independent mechanisms. We have now clarified this observation in the revised manuscript (page 7). This observation aligns with previous studies reporting alternative pathways by which EGFR inhibits MHC-I antigen presentation (Wang et al., 2025, Cell Death Dis; Watanabe et al., 2019, Cancer Sci; Brea et al., 2016, Cancer Immunol Res).

4. In Figures 1F and 1G, the authors demonstrate that GLDC knockdown leads to increased H-2Kb /Db expression and reduced tumor cell viability in vitro. However, to strengthen the translational relevance of these findings and ensure consistency with the in vitro co-culture results, the authors should include data on tumor growth from an in vivo xenograft model. This would provide crucial validation that the observed effects of GLDC knockdown on antigen presentation and tumor suppression extend beyond the in vitro setting.

Reply: Following the reviewer's suggestion, we have now added the tumor growth and tumor weight data (see new Appendix Fig S2A). The results showed that tumor growth in mice subcutaneously injected with control cells was markedly faster than those injected with GLDC-deficient cells.

5. In Figures 2I and 7F, the authors present a single representative image of IHC staining. To improve the rigor and interpretability of these results, the authors should provide quantitative analysis of the IHC data, such as positive cell count, along with data from multiple biological replicates.

Reply: Following the reviewer's suggestion, we have now added the quantitative analysis of the IHC data and provided the data from multiple biological replicates (see new Figure 2I and Figure 7F-G).

Minor comments

1. *The authors should correct typographical errors found in lines 219, 393, 653, and 709—for example, "lessor decree" should be revised to "lesser degree."*

Reply: We have now corrected them.

2. *The authors have not clearly described the methodology used for the CRISPR screening. Details such as the library used, screening conditions, selection criteria, and analysis pipeline are essential for evaluating the validity and reproducibility of the findings. We recommend the authors provide a more comprehensive explanation of how the screening was performed, either in the main text or supplementary materials.*

Reply: We have now added the methodology used for the CRISPR screening (page 34).

3. *In Figure 1A, the x-axis of the histogram lacks a scale, making it difficult to interpret the data accurately. The authors are advised to include appropriate axis labels and units.*

4. *In Figure 1A, please include the p-values for the CRISPR screening results. This could be visualized in the form of a volcano plot, for example.*

Reply: We have now presented the data in a volcano plot to clearly present the data.

5. *The authors use varying labels for the fluorescence intensity, "MFI," and "relative MFI," throughout the manuscript. For clarity and consistency, we recommend standardizing the terminology across all text and figure panels. This will help readers to better follow the data presentation and interpretation.*

Reply: We have now standardized the terminology across all text and figure panels as "relative MFI".

6. *Regarding Figure 5, please clarify in the results description that SMARCE1 is referred to as "CE1" in the figure.*

Reply: We have now changed “CE1” to “SMARCE1” throughout the text and figures.

Referee #3:

1. The rationale for using the H1299 cell line in the initial CRISPR screen should be provided, particularly given the focus on MHC-I expression.

Reply: H1299 and A549 cells exhibited higher constitutive MHC-I expression compared to PC9 cells. We performed a genome-wide sgRNA screen targeting 107 metabolic enzymes. To optimize screening efficiency, we selected H1299 over A549 based on higher infectivity for sgRNA transduction and greater suitability for post-infection candidate selection. The experimental rationale has been clarified in the revised manuscript (page 6).

2. Additional hits besides GLDC are evident in Figure 1A. The authors should acknowledge this and justify their decision to focus on GLDC.

Reply: Our initial CRISPR screen identified 4 metabolic enzymes including GLDC that had the most dramatic effects on MHC-I expression (Figure 1A). Subsequent validation experiments revealed that GLDC knockout produced the most pronounced increase in the surface expression of MHC-I (see new Appendix Fig S1A). Notably, knockout of the other 3 genes significantly compromised cellular viability, suggesting their essential roles in fundamental metabolic process. Together, we prioritized GLDC for further investigation.

3. In Figure 1C, Afatinib appears to further increase B2M levels in gGLDC compared to gNC cells. Please recheck the statistical analysis.

Reply: Our reanalysis confirmed that GLDC-deficiency modestly increased HLA-ABC and B2M expression in Afatinib-treated cells (Fig 1C). EGFR inhibition impaired the up-regulation of MHC-I induced by GLDC-deficiency, but it did not completely abolish this effect. We have now corrected it in the revised manuscript (page 7).

Based on our results, EGFR activation triggers SRC-mediated phosphorylation and nuclear translocation of GLDC, leading to transcriptional inhibition of the downstream MHC-I genes. However, EGFR represents one upstream regulator of SRC. Afatinib treatment inhibits EGFR signaling but may not block SRC activity. This may have led to GLDC-deficiency weakly up-regulated the MHC-I levels in Afatinib-treated cells.

4. Tumor size and weight data should be included for completeness in Figure 1F.

Reply: We have now added the tumor growth and tumor weight data (see new Appendix Fig S2A).

5. In Figure 1H, clarify whether human and mouse GLDC proteins are conserved

sufficiently to justify the use of human GLDC to rescue mouse GLDC knockout.

Reply: In GLDC-knockout A549 and H1299 cells, we expressed human GLDC variants (wild-type, G763D, or G768E). For mouse CT26- and MC38-EGFR^{ex19del} cells, we introduced mouse GLDC variants (wild-type, G768D, or G773E). We have now clarified it in the revised manuscript (page 8 and 46).

6. In Figure 2G, tumors from WT-GLDC rescue groups should also be included for proper comparison.

Reply: We have now added the WT-GLDC rescue group in Figure 2G.

7. Specify in Figures 2I-J whether CD8⁺ cells were sorted from CD45⁺ populations.

Reply: The CD8⁺ T cells were sorted from CD45⁺ populations. We have now clarified it in the revised manuscript (page 48).

8. In Figures 3K-L, it should be clarified whether GLDC directly binds to STAT1 or chromatin.

Reply: ChIP experiments demonstrated that GLDC could not directly bind to the promoter region of *IRF1* gene in control or STAT1-deficient cells (see new Appendix Fig S4I).

In Figure 5I and Appendix Figure S9B, endogenous co-immunoprecipitation experiments showed that GLDC-STAT1 association required SMARCE1, as this interaction was abolished in SMARCE1-deficient cells. In addition, SRC-mediated phosphorylation of SMARCE1^{Y170} was required for the interaction of SMARCE1 with GLDC, and SMARCE1^{Y170F} failed to interact with GLDC (Appendix Fig S8I and J). I206/L207/V218 of SMARCE1 was important for its interaction with STAT1, and SMARCE1^{I206A/L207A/V218A} (SMARCE1^{ILV-A}) failed to interact with STAT1 (Appendix Fig S9E and F). To validate these findings, we reconstituted SMARCE1-knockout A549 cells with wild-type SMARCE1, SMARCE1^{Y170F}, SMARCE1^{ILV-A} or SMARCE1^{Y-F/ILV-A}. Co-immunoprecipitation experiments confirmed that GLDC was associated with STAT1 in wild-type SMARCE1-reconstituted cells but not in SMARCE1^{Y170F}, SMARCE1^{ILV-A} or (SMARCE1^{Y-F/ILV-A})-reconstituted cells (Fig 5J). These results suggest that GLDC interacts with STAT1 through SMARCE1.

9. Total GLDC blots are missing in Figures 4D, 4E, and 4G and should be included.

Reply: We have now added them.

10. An HA blot in the HA-IP sample is missing in Figure 4I and needs to be added.

Reply: We have now added it.

11. Figure 4 suggests that tyrosine-phosphorylated GLDC blocks STAT1 binding to MHC-I gene promoters. Clarify whether this is due to increased affinity of GLDC for the promoters, or mediated through other transcription factors.

Reply: ChIP experiments demonstrated that GLDC could not directly bind to the promoter region of *IRF1* gene in control or STAT1-deficient cells (see new Appendix Fig S4I). As shown in Figure 5, SMARCE1 was a co-activator of STAT1, and nuclear GLDC inhibited STAT1-triggered MHC-I expression by hijacking SMARCE1.

12. The rationale for using HEK293 cells in Table EV3, rather than the tumor lines used in earlier figures, should be explained. Additionally, since the function appears nucleus-specific, a nuclear GLDC interactome would be more appropriate.

Reply: As shown in Fig 4 and Appendix Fig S5-6, EGFR activation triggers SRC-mediated phosphorylation and nuclear translocation of GLDC. This mechanism was consistently observed across cell systems, including HEK293 cells. We selected HEK293 cells for large-scale GLDC expression and mass spectrometry analysis due to their superior transfection efficiency compared to A549 or H1299 cells, enabling robust protein yield for proteomic studies.

According to our results, the function of GLDC to regulate MHC-I expression is nucleus-specific (Fig 4L-M). The nuclear GLDC interactome is indeed more appropriate, but our current data are sufficient to support our conclusions.

13. In Figure EV6C, provide evidence that there are at least two pools of SMARCE1—one associated with SWI/SNF complexes and another with phosphorylated GLDC. Also, indicate whether GLDC remains phosphorylated upon nuclear translocation.

Reply: As shown in Figure EV6C (now Appendix Fig S7D), GLDC was only associated with SMARCE1 but not the other core components of SWI/SNF complex including SMARCB1, SMARCD1, SMARCA4, BRD7, BRD9 and ACTL6A.

EGF treatment or overexpression of SRC induced the interaction of SMARCE1 with wild-type GLDC but not GLDC^{Y993F/Y1008F} or GLDC^{L998A/V999A} (Appendix Fig S7A and B). Endogenous co-immunoprecipitation experiments further indicated that EGF induced the interaction of GLDC with SMARCE1 in the nucleus (Fig 5A). These results suggest that SMARCE1 is associated with GLDC after EGFR activation.

To investigate whether SMARCE1 is associated with SWI/SNF complexes after

EGFR activation, we performed co-immunoprecipitation experiments. The results showed that SMARCE1 was associated with the core components of SWI/SNF complex, EGF stimulation did not affect their interaction (see new Appendix Fig S7C). These results suggest that SMARCE1 is constitutively associated with SWI/SNF complexes before and after EGFR activation.

In addition, immunoblotting experiments indicated that GLDC remained phosphorylated upon nuclear translocation (see new Appendix Fig S5J).

14. From the ChIP assays, clarify whether pSMARCE1^{Y170} co-localizes with GLDC at MHC-I promoters, or whether their interaction occurs primarily in chromatin-free nuclear fractions.

Reply: ChIP experiments demonstrated that GLDC could not directly bind to the promoter region of *IRF1* gene either in control or STAT1-deficient cells (see new Appendix Figure S4I). Furthermore, SMARCE1 bound to the *IRF1* promoter in SMARCE1^{Y170F}-reconstituted cells but not in wild-type SMARCE1-, SMARCE1^{ILV-A}- or SMARCE1^{Y-F/ILV-A}-reconstituted cells (Fig 5M). While GLDC-deficiency promoted SMARCE1 binding to *IRF1* promoter region, but this effect was abolished in STAT1-deficient cells (Fig 5N). These results suggest that SMARCE1 binds to the promoter region of *IRF1* gene through STAT1, and this interaction is inhibited by nuclear GLDC-SMARCE1 interaction. In addition, SMARCE1^{ILV-A} failed to interact with STAT1, but it interacted with GLDC (Appendix Fig S9G). Taken together, the interaction of SMARCE1 with GLDC does not occur primarily at MHC-I promoters.

15. Further clarification is needed on how DMAP1 and SMARCE1 binding to GLDC influence each other, as shown in Figure 6.

Reply: As shown in new Appendix Fig S10L and M, DMAP1-deficiency did not affect the interaction of GLDC with SMARCE1, and the interaction between GLDC and DMAP1 was also not affected by SMARCE1-deficiency.

16. In Figure 6K, the authors should speculate on why CpG methylation does not appear to impair STAT1 binding to the promoters in this context.

Reply: DNA methylation suppresses gene transcription through two main mechanisms, directly interfering with the binding of transcription factors to gene promoters and indirect recruitment of transcriptional repressors by 5mC-binding proteins. As shown in Figure 6K, the binding of STAT1 to *IRF1* promoter region was unaffected in DMAP1 or DNMT1 knockout cells in comparison to control cells, suggesting that the DMAP1/DNMT1-induced promoter DNA hypermethylation does not impair STAT1 binding to downstream target gene promoter. This observation aligns with previous studies that DNMT1 and DNMT3b cooperate to mediate the

IRF8 promoter DNA methylation, and the DNA methylation inhibits STAT1 activity without blocking its binding to the GAS element of *IRF8* promoter (McGough et al., 2008, Cancer Research). Further mechanisms reveal that *IRF8* promoter methylation enables methyl-CpG binding domain protein 1 (MBD1) to recruit repressor PIAS1, thereby inhibiting STAT1 function (McGough et al., 2008, Cancer Research). We have now added these mechanistic insights in the revised manuscript (page 29).

17. In Figure 6N, it remains unclear how SMARCE1 functions in DMAP1-altered cells. It should also be explored whether DMAP1 induces global CpG methylation changes that secondarily impact MHC-I promoter activity.

Reply: In this study, we show that SMARCE1 functions as a co-activator of STAT1 that promotes STAT1 binding to DNA promoter regions through direct interaction. We also reveal that GLDC suppresses STAT1-mediated MHC-1 expression by binding to SMARCE1. Key findings revealing whether DMAP1 affects this mechanism include: immunoblotting experiments indicated that DMAP1-deficiency did not affect the interaction of SMARCE1 with GLDC or STAT1 (see new Appendix Fig S10M); ChIP experiments showed that DMAP1-deficiency did not impair the binding of SMARCE1 or STAT1 to *IRF1* promoter region (see new Appendix Fig S10J and K). These results suggest that the function of SMARCE1 as a co-activator of STAT1 is not affected in DMAP1-deficient cells.

While previous studies have demonstrated that global CpG methylation negatively regulates MHC gene expression (Pappalardi et al., 2021, Nature Cancer; Luo et al., 2018, Nature Communications; Deshmukh et al., 2025, Clinical Epigenetics), our work specifically elucidates the GLDC-SMARCE1-STAT1 axis in MHC-I control. Whether DMAP1 induces global CpG methylation changes that secondarily impact MHC-I promoter activity needs further investigation in the future.

Dear Dr Li,

Thank you for submitting your revised manuscript (EMBOJ-2025-120829R) to The EMBO Journal, as well for your patience with our feedback. Your amended study was sent back to the referees for their scientific reassessment, and we have received reports from two of them, which I enclose below. Please note that while referee #1 was not able at this time to reassess your amended study we have considered your response to the issues raised by this expert editorially and found these to be addressed satisfactorily. As you will see, the other reviewers state that the work has been substantially enhanced by the revisions and they are now broadly in favour of publication.

Thus, we are pleased to inform you that your manuscript has been accepted in principle for publication in The EMBO Journal.

We now need you to take care of a number of issues related to formatting and data presentation as detailed below, which should be addressed at re-submission.

Please contact me at any time if you have additional questions related to below points.

As you might have seen on our web page, every paper at the EMBO Journal now includes a 'Synopsis', displayed on the html and freely accessible to all readers. The synopsis includes a 'model' figure as well as 2-5 one-short-sentence bullet points that summarize the article. I would appreciate if you could provide this figure and the bullet points.

Thank you for giving us the chance to consider your manuscript for The EMBO Journal. I look forward to your final revision.

Again, please contact me at any time if you need any help or have further questions.

Best regards,

Daniel Klimmeck

>> Please limit the keywords to your study to maximally five.

>> Provide a completed Author Checklist.

>> Author Contributions: Remove the author contributions information from the manuscript text. Note that CRediT has replaced the traditional author contributions section as of now because it offers a systematic machine-readable author contributions format that allows for more effective research assessment. and use the free text boxes beneath each contributing author's name to add specific details on the author's contribution.

More information is available in our guide to authors.
<https://www.embopress.org/page/journal/14602075/authorguide>

>> Adjust the title of the 'Competing Interests' section to 'Disclosure and Competing Interests Statement' and move after Acknowledgements.

>> Section order should be corrected as follows: title page with complete author information, abstract, keywords, introduction, results, discussion, methods, data availability section, acknowledgements, disclosure and competing interests statement, references, main figure legends, tables, expanded figure legends.

>> Figures in separate files: Please upload the main figures and Figure EV1 as high-resolution individual figure files, in PDF, TIFF or EPS format. Powerpoint format is not accepted.

>>Funding: Please add the 'National Natural Science Foundation of China (32188101 and 32300758), the Fundamental Research Funds for the Central Universities (2042022dx0003, 2042025kf0043), the Natural Science Foundation of Wuhan (2024040701010031), and the China Postdoctoral Science Foundation (2022TQ0236, 2023M732694)' as funders in our online

system.

*Appendix file with ToC: Please add a table of contents to the file with the supplementary image. Please include page numbers. The legend for Appendix Figure S8 is included twice, please delete the duplicate version. Please save the file in PDF format and add the appendix tables to the PDF and the table of contents. If it is difficult to fit Appendix Table S3 into the appendix PDF, please remove it from the appendix and rename it Table EV1.

>> Reagents and Tools table: Please upload as a separate file using the existing template in the Guide For Authors, listing key reagents, experimental models, software and relevant equipment.

>> Data availability section: please indicate: 'No data amenable to large-data repository deposition have been generated in this study.' .

>> Consider additional changes and comments from our production team as indicated below:

- Figure legends:

Please define the annotated p values ****/**/**/* as well as provide the exact p-values for the same in the legend of figure 1B-H; 2A-K; 3D, E, F, G, H, J, K, L; 4M, 5B-E, F, G, K-P; 6A-N; 7B, D, E, G-K; as appropriate.

Referee #2:

The authors addressed my concerns and this revised manuscript is ready to be published!

Referee #3:

The authors have addressed my raised concerns. I support the publication of the revised version.

The authors addressed the remaining editorial issues.

Dear Dr Li,

Thank you for submitting the revised version of your manuscript. I have now evaluated your amended manuscript and concluded that the remaining minor concerns have been sufficiently addressed.

I am thus pleased to inform you that your manuscript has been accepted for publication in the EMBO Journal.

On a different note, I would like to alert you that EMBO Press offers a format for a video-synopsis of work published with us, which essentially is a short, author-generated film explaining the core findings in hand drawings, and, as we believe, can be very useful to increase visibility of the work. Please see the following link for representative examples and their integration into the article web page:

<https://www.embopress.org/doi/full/10.15252/emj.2019103932>

Best regards,

Daniel Klimmeck

Daniel Klimmeck, PhD
Senior Editor
The EMBO Journal
EMBO
Postfach 1022-40
Meyerhofstrasse 1
D-69117 Heidelberg
contact@embojournal.org
